# Catalytically inactive long prokaryotic Argonaute systems employ distinct effectors to confer immunity via abortive infection

Xinmi Song[1,4], Sheng Lei[1,4], Shunhang Liu[1,4], Yanqiu Liu[1,4], Pan Fu[1], Zhifeng Zeng[1], Ke Yang[1], Yu Chen[1], Ming Li [2], Qunxin She [3] & Wenyuan Han [1] ✉

Argonaute proteins (Agos) bind short nucleic acids as guides and are directed by them to recognize target complementary nucleic acids. Diverse prokaryotic Agos (pAgos) play potential functions in microbial defense. The functions and mechanisms of a group of full-length yet catalytically inactive pAgos, long-B pAgos, remain unclear. Here, we show that most long-B pAgos are functionally connected with distinct associated proteins, including nucleases, Sir2-domain-containing proteins and trans-membrane proteins, respectively. The long-B pAgo-nuclease system (BPAN) is activated by guide RNA-directed target DNA recognition and performs collateral DNA degradation in vitro. In vivo, the system mediates genomic DNA degradation after sensing invading plasmid, which kills the infected cells and results in the depletion of the invader from the cell population. Together, the BPAN system provides immunoprotection via abortive infection. Our data also suggest that the defense strategy is employed by other long-B pAgos equipped with distinct associated proteins.

Argonaute proteins (Agos) are important defense elements in both eukaryotes and prokaryotes[1,2]. Eukaryotic Agos (eAgos) provide immunity against viruses and transposons by RNA silencing[3,4]. In the RNA silencing pathways, eAgos are loaded with short RNA fragments (guides) generated from viruses, transposons or genomic transcripts, and are directed to recognize and/or cleave target RNA for silencing. The structural basis of the RNA-guided RNA recognition of eAgos has been revealed[5]. They consist of two lobes, which comprise N-terminal (N) and PAZ (PIWI–Argonaute–Zwille) domains, and MID (Middle) and PIWI (P-element Induced Wimpy Testis) domains, respectively. The MID domain contains a pocket that binds to the 5′-end of the guide RNA, while the PAZ domain anchors the 3′-end of the guide. The PIWI domain of eAgos forms an RNaseH fold and usually contains a DEDX (where X denotes D, H, or K) catalytic tetrad, which is essential for RNA cleavage activity.

Bioinformatics analyses reveal that prokaryotic Agos (pAgos) are more diverse than eAgos[2,6,7]. Many pAgos show distinct domain architectures[2,6]. Genomic context studies identify that the neighborhood of *pAgo* genes encodes several types of the so-called pAgo-associated proteins, which were predicted to be functionally linked with pAgos[2,6]. A more comprehensive study has classified pAgos into three groups according to the phylogenetic analyses: long-A, long-B and short pAgos[7]. Long-A and long-B pAgos contain all four domains as eAgos, while short pAgos lack the N and PAZ domains. Most of the long-A pAgos are active nucleases since their catalytic tetrad is intact. Many long-A pAgos are directed by ssDNA guides to bind and cleave ssDNA targets[8–13]. In vivo, their guides are more often derived from extrachromosomal genetic elements and/or multicopy genetic elements and thus they can be directed to target invading viruses and plasmids[8,13]. Long-A pAgos also acquire guides from the region of

[1]State Key Laboratory of Agricultural Microbiology and College of Life Science and Technology, Hubei Hongshan Laboratory, Huazhong Agricultural University, 430070 Wuhan, China. [2]CAS Key Laboratory of Microbial Physiological and Metabolic Engineering, State Key Laboratory of Microbial Resources, Institute of Microbiology, Chinese Academy of Sciences, Beijing, China. [3]CRISPR and Archaea Biology Research Center, State Key Laboratory of Microbial Technology, Shandong University, Binhai Road 72, 266237 Jimo, Qingdao, China. [4]These authors contributed equally: Xinmi Song, Sheng Lei, Shunhang Liu, Yanqiu Liu. ✉e-mail: hanwenyuan@mail.hzau.edu.cn

replication termination and can be directed by them to resolve replicated DNA molecules[13,14]. Other long-A pAgos show diverse guides and target preferences in vitro[15–19], suggesting that they may play versatile physiological functions.

Short pAgos are catalytically inactive as a result of the mutation of the catalytic tetrad, and their functions have been a mystery for a long time until most recently. It was reported that short pAgos from many bacteria and a (pseudo)short pAgo from *Sulfolobus islandicus* (Si) constitute defense systems together with their associated proteins[20–24]. These short pAgos systems confer immunity against viruses and plasmids via an abortive infection (Abi) response[25], a defense strategy that kills the infected cells or induces cell dormancy to suppress the spreading of the invaders[26].

The long-B pAgo group is featured with the four-domain architecture yet deactivated nuclease activity[7]. A representative long-B pAgo, *Rhodobacter sphaeroides* (Rs) Ago, binds short RNA as guides and is directed to recognize target DNA[27,28]. The DNA binding results in suppression of plasmid-encoded gene expression and/or plasmid degradation in a heterologous host[27,28]. In addition, the guide/target-binding mode of RsAgo is different from that of typical long-A pAgos, possibly due to the structural variations within the N and PAZ domains[28]. Moreover, the gene neighborhood of long-B pAgos also encodes pAgo-associated proteins[7], as exemplified by RsAgo, the downstream gene of which encodes a predicted nuclease[27]. However, how these proteins could be functionally connected with their respective long-B pAgos remains unknown.

In this study, we explored the diversity of long-B pAgos and found that long-B pAgos are genetically and functionally associated with nucleases, NADases and trans-membrane proteins. The long-B pAgo-nuclease system performs RNA-guided unspecific DNA degradation following recognition of a specific DNA target, and relying on the activity, the system mediates an Abi response to provide immunoprotection against invading plasmid. The findings indicate that both CRISPR-Cas systems and pAgo systems can utilize the target recognition-activated collateral DNA degradation as a defense strategy[29–32] and provide novel opportunities to develop pAgo-based biotechnological applications.

## Results

### Long-B pAgos cluster with potential toxic effectors
Bioinformatic analysis indicates that long-B pAgos tend to associate with a number of proteins that can be clustered into several orthogroups (og)[7]. Among them, og_15 (nuclease), og_44 (SIR2_2 domain-containing protein) and og_100 (protein with unknown domain) are usually encoded by simple operons that only contain 2 or 3 genes[7]. To further analyze the connection between long-B pAgos and their associated proteins, we constructed a phylogenetic tree of long-B pAgos and marked them with different colors according to their associated proteins (Fig. 1a, Supplementary Fig. 1). This reveals that the long-B pAgos associated with og_15, og_44, og_54 (VirE N-terminal domain-containing protein) and og_100 can be clustered into separated subclades. Further, a comparison of the phylogenetic trees of og_15, og_44 and og_100, and their respective pAgos suggests the coevolution of the long-B pAgos and the associated proteins (Fig. 1b–d). In addition, the operons encoding og_15, og_44 and og_100 are organized in the same structure, with *pAgos* upstream of their associated genes (Fig. 1e). Together, the analyses strongly suggest that og_15, og_44 and og_100 are functionally connected with their respective long-B pAgos.

To gain an insight into the possible function of og_100 that was annotated as "protein with unknown domain"[7], we predicted the structure of og_100 members using DeepTMHMM and Alphafold2. This reveals that the og_100 members are trans-membrane (TM) proteins (an example shown in Supplementary Fig. 2g, h). Nucleases, SIR2-like proteins and TM proteins are widely found in Abi defense

systems[33–39]. Their association with long-B pAgos implies that the related long-B pAgo systems might also mediate Abi responses. Based on the nomenclature proposed for short pAgo systems and SiAgo system[21,24], the long-B systems are named BPAN (long-B prokaryotic Argonaute nuclease), BPAS (long-B prokaryotic Argonaute Sir2) and BPAM (long-B prokaryotic Argonaute trans-membrane) systems, respectively. Correspondingly, the associated proteins are named bAgaN (long-B pAgo-associated nuclease), bAgaS (long-B pAgo-associated Sir2) and bAgaM (long-B pAgo-associated trans-membrane). To reveal the functions and molecular mechanisms of these long-B pAgo systems, we selected the *Escherichia coli* (Ec) BPAN system, a BPAS system from unclassified *Gammaproteobacteria bacterium* (Gb) and the *Elizabethkingia anophelis* (Ea) BPAM system for analysis (Fig. 1e and Supplementary Data 1).

### bAgaN is a PD-(D/E)XK superfamily DNase
We began by focusing on the biochemical properties of bAgaN. bAgaN was annotated as a member of the RecB-like protein family, belonging to the PD-(D/E)XK superfamily[7]. Members of the superfamily are involved in restriction-modification systems, DNA metabolism, tRNA splicing[40]. Recently, many members of the superfamily have been found as effectors of type III CRISPR-Cas and CBASS defense systems[30,31]. We expressed EcbAgaN in *E. coli* BL21 and purified it to apparent homogeneity (Supplementary Fig. 4a). The analysis by multi-angle light scattering coupled with size exclusion chromatography (SEC-MALS) indicates that EcbAgaN forms a dimer in solution (Fig. 2a). We predicted the dimer structure of EcbAgaN (Supplementary Fig. 2a). The analysis reveals that EcbAgaN is composed of two domains that are connected by an unstructured link and the C-terminal domain adopts a restriction endonuclease-like fold (Supplementary Fig. 2a, b), which possesses the conserved D-EVK motif as shown by the sequence alignment analysis (Supplementary Fig. 3a). A database search using the DALI structure-comparison server[41] reveal that EcbAgaN has structural similarity to the type III CRISPR-Cas system accessory nuclease Card1 (PDB: 6wxx)[32] and Can2 (PDB: 7bdv)[42], and the type IIS restriction endonuclease R.BspD6I (PDB: 2ewf)[43]. Then, we analyzed the nuclease activity of EcbAgaN with various substrates. The results show that EcbAgaN efficiently cleaves ssDNA, dsDNA and ssDNA from a DNA/RNA duplex but shows no activity toward RNA (Fig. 2b), and the DNase activity of EcbAgaN is dependent on $Mn^{2+}$ (Supplementary Fig. 4b). The cleavage of dsDNA and ssDNA generates short oligonucleotide products of various sizes, most of which are smaller than 16 nt (Supplementary Fig. 4c). We then constructed the mutants that carry alanine substitutions of the D-EVK motif (M1: D298A; M2: E309A-K311A). Analysis of the mutants reveals that they are inactive for DNA cleavage (Fig. 2c), indicating that the motif is essential for the nuclease activity. In addition, EcbAgaN also efficiently degrades plasmid and genomic DNA that are extracted from *E. coli* DH5α cells (Fig. 2d), suggesting that EcbAgaN can cleave methylated DNA. Further, the nuclease activity of EcbAgaN is inhibited by moderate concentrations of NaCl and KCl in vitro (Supplementary Fig. 4d).

### The EcBPAN system induces cell death via degrading genomic DNA
We noticed that EcAgo and its close homologs have two predicted starting codons, encoding a 743 amino-acids (a.a.) protein and a 731-a.a. Ago respectively (Supplementary Fig. 5a). Analysis of the structures of the two proteins reveals that the N-terminal 12 a.a. in the 743 a.a protein is disordered and absent from RsAgo (Supplementary Fig. 5b, c), indicating that the 731 a.a version is the correct form of the EcAgo protein (hereafter EcAgo). We constructed strains expressing EcAgo and EcbAgaN individually or both of them (i.e., the EcBPAN system) using the expression plasmids pCDF-EcAgo and pET28T-EcbAgaN (Supplementary Data 2). The strains, as well as the control strain containing empty vectors (EV), were grown in the presence of

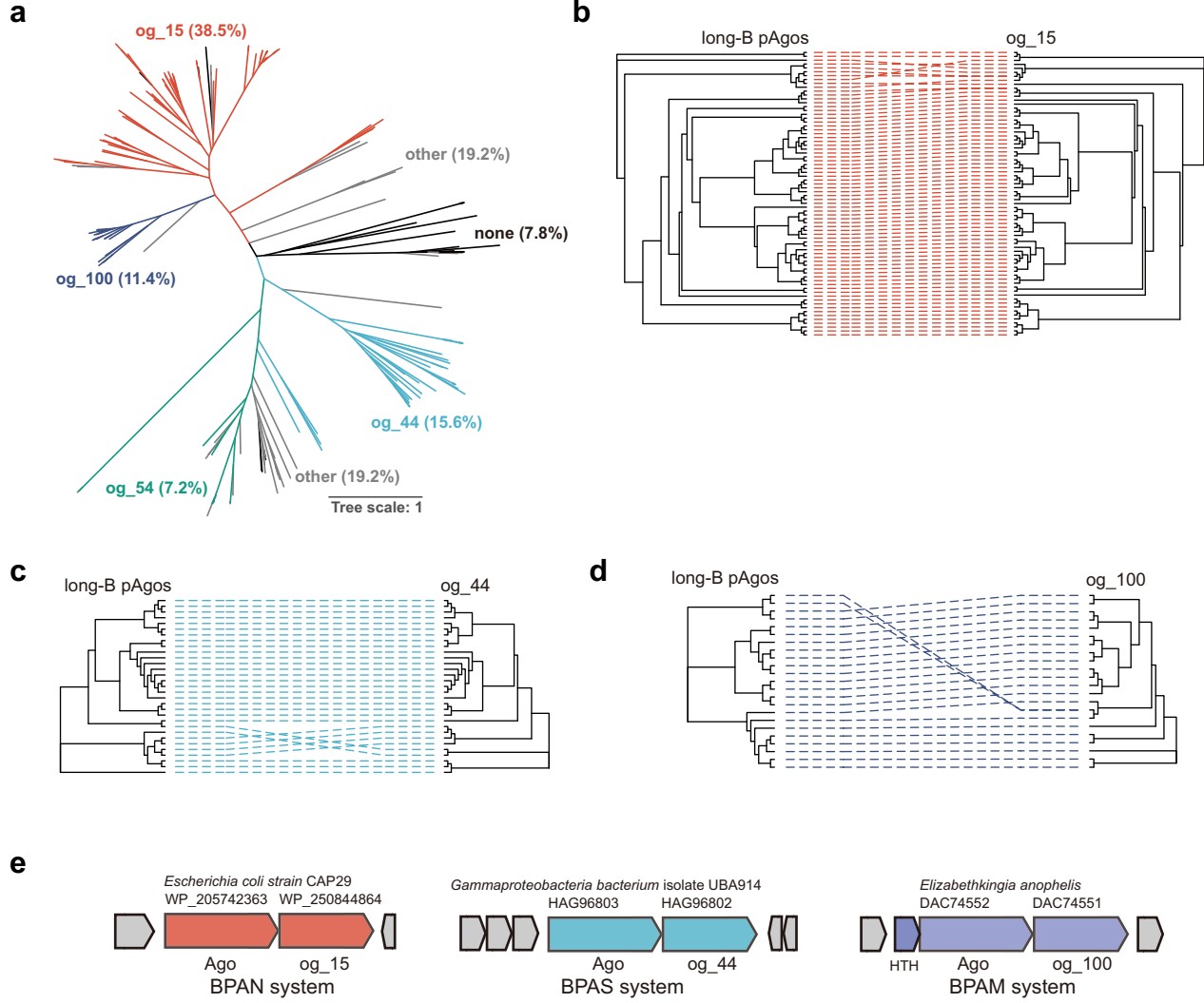

**Fig. 1 | Phylogenetic analysis of long-B pAgo systems. a** Maximum likelihood-based unrooted phylogenetic tree of long-B pAgos. The proteins were derived from the supplementary data provided by Ryazansky et al.[7]. The long-B pAgos associated with different orthogroups (ogs) are marked with different color, and the corresponding ogs are indicated. other: the associated ogs do not belong to og_15, og_44, og_54 or og_100; none: none of the ogs are encoded by the adjacent genes or the adjacent genes do not form an operon-like structure with the long-B *pAgo* genes.

The percentage of each subclade is shown. **b–d** Tanglegram of phylogenetic trees of og_15, og_44 and og_100 (right), and their respective long-B pAgos (left). **e** Operon structure and neighborhood genes (shown in gray) of representative long-B pAgo systems. The names of the systems, their source organisms, and the encoded proteins and their accession numbers are indicated. HTH, putative helix-turn-helix transcription regulator.

inducers (IPTG and aTc) but without antibiotics. At 2 h during the induction, cell viability was assessed by plating the cells onto plates with or without antibiotics. The results derived from the antibiotic-free plates show that individual expression of EcAgo and EcbAgaN has little effect on the cell viability (Fig. 3a), in line with that EcbAgaN can be inhibited by physiological salt concentrations (Supplementary Fig. 4d). In comparison, expression of the EcBPAN system results in an about 279-fold reduction in the cell viability (Fig. 3a). The results indicate that EcAgo and EcbAgaN cooperatively induce cell death. In addition, we constructed the mutated EcBPAN systems containing EcbAgaN dead mutants (M1 and M2). Western blot analysis of the wild type and mutated EcbAgaN proteins reveals that neither mutation affects the stability of EcbAgaN (Supplementary Fig. 6a). The mutated EcBPAN systems do not induce any reduction in the cell viability, indicating that the cell death is dependent on the nuclease activity of EcbAgaN.

To reveal how the EcBPAN system induces cell death, we analyzed the genome integrity of the host cells. Cell samples were taken from the cultures during the induction and subjected to genomic DNA extraction and DAPI-staining. Gel electrophoresis analysis of the

genomic DNA indicates that the EcBPAN system results in extensive genomic DNA degradation (Fig. 3b). Meanwhile, analysis of the DAPI-stained cells with flow cytometry shows that the EcBPAN system induces a dramatic decrease of the cellular DNA content (Fig. 3c). By comparison, the dead mutations of EcbAgaN abolish the observed genomic DNA degradation (Fig. 3b, c). Together, these data indicate that EcAgo and EcbAgaN cooperatively induce cell death by degrading genomic DNA via the nuclease activity of EcbAgaN.

## The CloDF13 origin is the trigger of the EcBPAN system

Next, we compared the cell viability on the antibiotic-free plates and the plates containing Kanamycin (Kan, for selection of pET28aT-EcbAgaN) or streptomycin (Str, for selection of pCDF-EcAgo), or both of them (Fig. 3a). The results show that individual EcAgo or EcbAgaN, or the mutated EcBPAN systems do not lead to any plasmid depletion. By comparison, the wild-type EcBPAN system induces a 13-fold additional decrease in the cell viability on the Str plates compared to the antibiotic-free plates (Fig. 3a), indicative of the depletion of the pCDF-EcAgo plasmid. Nevertheless, the cell viability on the Kan plates is the

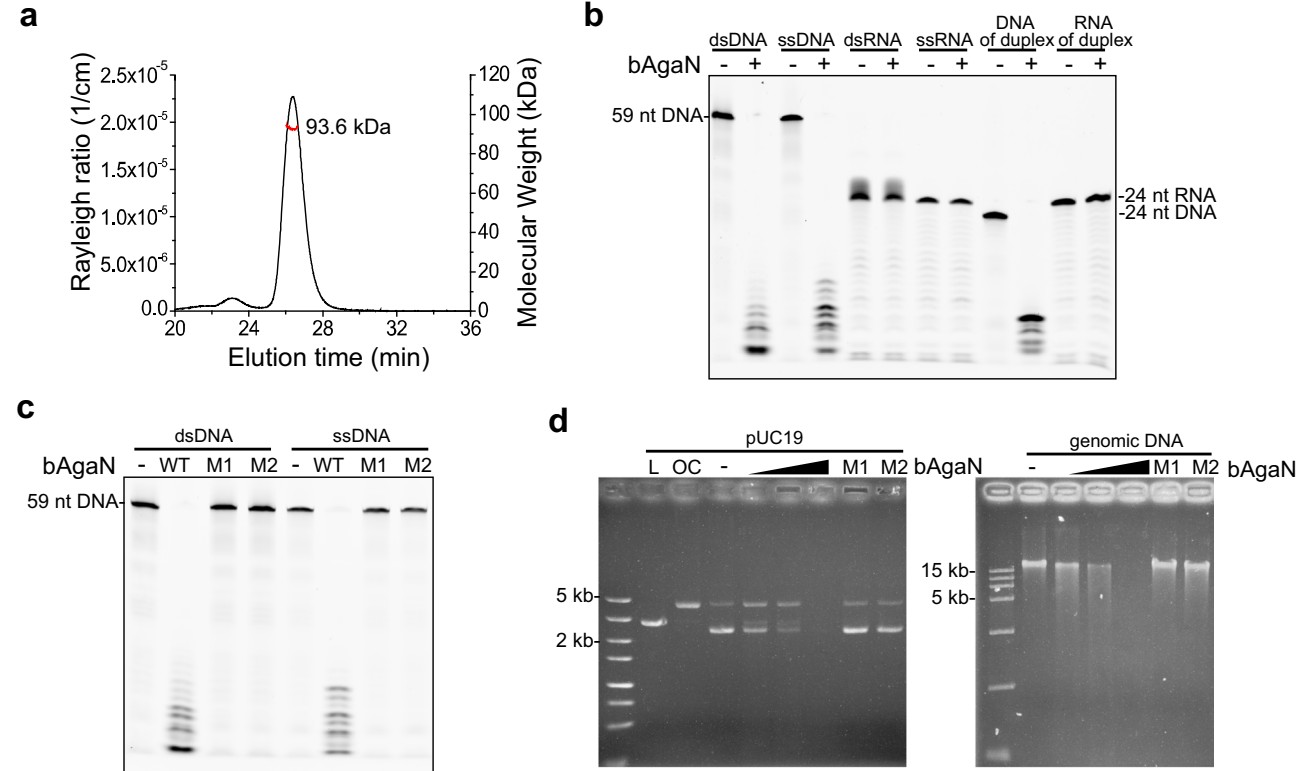

**Fig. 2 | EcbAgaN is a DNase. a** Multi-angle light scattering coupled with size exclusion chromatography (SEC-MALS) analysis of EcbAgaN. **b** Substrate specificity of EcbAgaN. FAM-labeled dsDNA, ssDNA, dsRNA, ssRNA and DNA/RNA duplexes that contained either FAM-labeled DNA strand or RNA strand were incubated with EcbAgaN and then analyzed by denaturing polyacrylamide gel electrophoresis.

**c** DNase activity of EcbAgaN and its mutants. M1: D298A; M2: E309A-K311A. **d** Degradation of plasmid DNA and genomic DNA by EcbAgaN and its mutants. Plasmid DNA (pUC19) and *E. coli* genomic DNA were incubated with a gradient of EcbAgaN and then analyzed by agarose gel electrophoresis. L, linear pUC19 DNA; OC, open circular pUC19 DNA. Source data are provided as a Source Data file.

same as that on the antibiotic-free plates. The results suggest that the EcBPAN system mediates the selected depletion of pCDF-EcAgo.

To explore the phenomenon, we expressed the two proteins using bacterial artificial chromosome (BAC) vector with araBAD promoters. In this case, induction of gene expression does not induce cell viability reduction (Fig. 3d). However, in the transformants carrying pCDF-EGFP, induction of gene expression results in ~46-fold reduction in cell viability on antibiotic-free plates, indicating that pCDF-EGFP activates EcBPAN to trigger cell death. Consistently, EcBPAN activation also results in extensive genomic DNA degradation, and both cell death and genomic DNA degradation are dependent on the nuclease activity of EcbAgaN (Supplementary Fig. 7a, b). In addition, EcBPAN activation also depletes pCDF-EGFP from the survivors (~3-fold cell viability reduction on Str plates compared to antibiotic-free plates) but not the BAC expression plasmid (Fig. 3d). The selective killing of pCDF-EGFP-carrying cells may contribute to the depletion of pCDF-EGFP from the cell population since the pCDF-EGFP-free cells would gain a growth advantage.

Then, we aim to analyze how pCDF-EGFP triggers the EcBPAN system. Since the CloDF13 origin and T7 expression cassettes have been shown to act as triggers for short pAgo systems[22,24], we replaced the CloDF13 origin of pCDF-EGFP with the ColA origin and the pSC101 replicon respectively and also removed the T7 expression cassettes (T7Es) of pCDF-EGFP. Then, the cell viability of the cells containing these variant plasmids was analyzed on the plates in the presence or absence of IPTG (Fig. 3e). The results indicate that the plasmids containing ColA origin or pSC101 replicon fail to trigger the EcBPAN system, while removal of T7Es does not affect EcBPAN-mediated cell death. Consistently, IPTG has little effect on cell death, either. The trigger of the EcBPAN system was further tested in the cells that

expressed the EcBPAN system from the pBAD24 plasmid. When expressed from pBAD24, the EcBPAN system does not trigger cell death unless a plasmid containing the CloDF13 origin is introduced (Supplementary Fig. 7c). Together, the data indicate that the CloDF13 origin activates the EcBPAN system to trigger cell death.

Next, we analyzed whether the EcBPAN system could provide immunoprotection against phages. The efficiency of plaque formation (EOP) of T5, T7 and lambda-vir on the bacterial lawn expressing the EcBPAN system was measured (Supplementary Fig. 7d). The results show that the EcBPAN system did not induce any reduction in the EOP of any phage.

### In vivo nucleic-acid binding of EcBPAN system

pAgo proteins are inherently directed by nucleic acid guides to recognize complementary nucleic acid targets[8,27], and the ability is employed by short pAgo and SiAgo systems to sense invaders[21,22,24]. To gain insight into how the EcBPAN system senses the invading plasmid, we analyzed the nucleic acids (NAs) associated with EcAgo in vivo. We purified EcAgo from the cells containing pBAD24-EcAgo-EcbAgaN and pCDF-EGFP and extracted the NAs from the protein sample. As control, the cells containing the pBAD24 empty vector (EV) and pCDF-EGFP were also subjected to protein purification and NA extraction. The NA samples were analyzed by Fast Thermo-sensitive Alkaline Phosphatase (FastAP) treatment, followed by T4 polynucleotide kinase (PNK) labeling with $\gamma P^{32}$-ATP. The results show that only the NAs that were extracted from EcAgo and treated with FastAP are visible after PNK labeling (Fig. 4a), indicating that the NAs are specifically associated with EcAgo and contain a 5′ phosphate group. Then, treatment of the labeled NAs by RNase and DNase respectively reveals that only RNase degraded the NAs

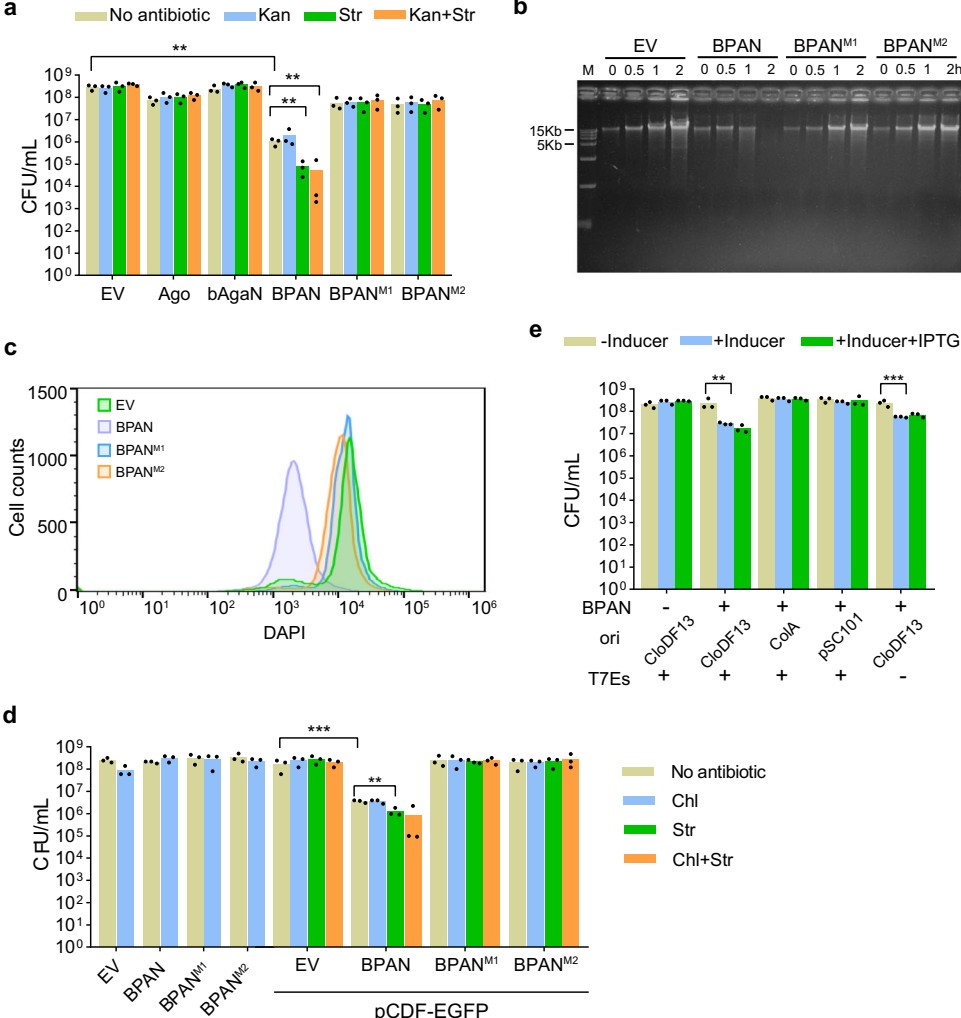

**Fig. 3 | EcBPAN system is activated by the CloDF13 origin and degrades genomic DNA. a** Expression of the EcBPAN system triggers cell death. The cells containing empty vector (EV), EcAgo, EcbAgaN, the EcBPAN system and the mutated EcBPAN systems (M1 and M2) were grown in LB medium supplemented with IPTG and aTc for 2 h. Then, the cells were plated onto the plates with or without antibiotics as indicated. The colony formation unit per mL (CFU/mL) was calculated. Data are presented as mean values with individual data points overlaid (*n* = 3 biological replicates). The *p*-values were calculated using a one-sided Student's *t*-test [**: *p* < 0.01; *p* = 0.001417, 0.005319, 0.005508 (from left to right)]. **b** EcBPAN system degrades genomic DNA in vivo. After the wild type and mutated EcBPAN systems were induced by IPTG and aTc, genomic DNA was extracted at indicated time points and analyzed by agarose gel electrophoresis. **c** Flow cytometry analysis of DNA content distributions in the cells after the wild type and mutated EcBPAN systems were induced by IPTG and aTc for 2 h. **d** The BAC (bacterial artificial chromosome)-expressed EcBPAN system is activated by pCDF-EGFP. The cells expressing wild-type and mutated EcBPAN systems (M1 and M2) in the

presence or absence of pCDF-EGFP were grown in LB medium supplemented with L-arabinose for 2 h. Then, the cells were plated onto the plates with or without antibiotics as indicated and CFU/mL was calculated. Data are presented as mean values with individual data points overlaid (*n* = 3 biological replicates). The *p*-values were calculated using one-sided Student's *t*-test [**: *p* < 0.01; ***: *p* < 0.001; *p* = 0.0005750, 0.004480 (from left to right)]. **e** The BAC-expressed EcBPAN system is activated by the CloDF13 origin. The cells carrying the EcBPAN system were transformed with pCDF-EGFP and its variants, where the CloDF13 origin was substituted with the indicated origins or the T7 expression cassettes (T7Es) were removed. Then, the cells were plated onto the plates with or without inducer, or with inducer and IPTG and CFU/mL was calculated. Data are presented as mean values with individual data points overlaid (*n* = 3 biological replicates). The *p*-values were calculated using one-sided Student's *t*-test [**: *p* < 0.01; ***: *p* < 0.001; *p* = 0.001178, 0.0008705 (from left to right)]. Inducer: L-arabinose. Source data are provided as a Source Data file.

(Fig. 4b). Together, the EcAgo-associated NAs are small RNAs containing a 5′ phosphate group.

Next, the extracted small RNAs were subjected to RNA sequencing, and meanwhile, the transcriptome of the corresponding culture was also analyzed. The results show that the abundance of small RNAs is correlated with the RNA sequences in the transcriptome (Fig. 4c, d, Supplementary Data 5), except that the plasmid-derived small RNAs are moderately enriched. Although the CloDF13 origin is the specific trigger of EcBPAN, only ~1% small RNAs are derived from the CloDF13 origin. This percentage is about 10 times of that of the ColE1 origin (0.11%), but much less than that of other genetic elements, e.g., the two

T7 expression cassettes (T7Es) of pCDF-EGFP (33.8%) and the whole pBAD plasmid (11.6%). The data suggest that EcAgo does not specifically acquire small RNAs from the CloDF13 origin-derived transcripts.

Then, we analyzed the length distribution and sequence bias properties of the small RNAs. The small RNAs are predominantly ~18–22 nt in length and show a bias toward A and U at the first two nt (Fig. 4e, f). The length distribution and sequence bias properties are applied to the small RNAs derived from both the CloDF13 origin and other genetic elements that do not activate the EcBPAN system, such as the pBAD24 plasmid and T7Es of pCDF-EGFP (Supplementary Figs. 10 and 11).

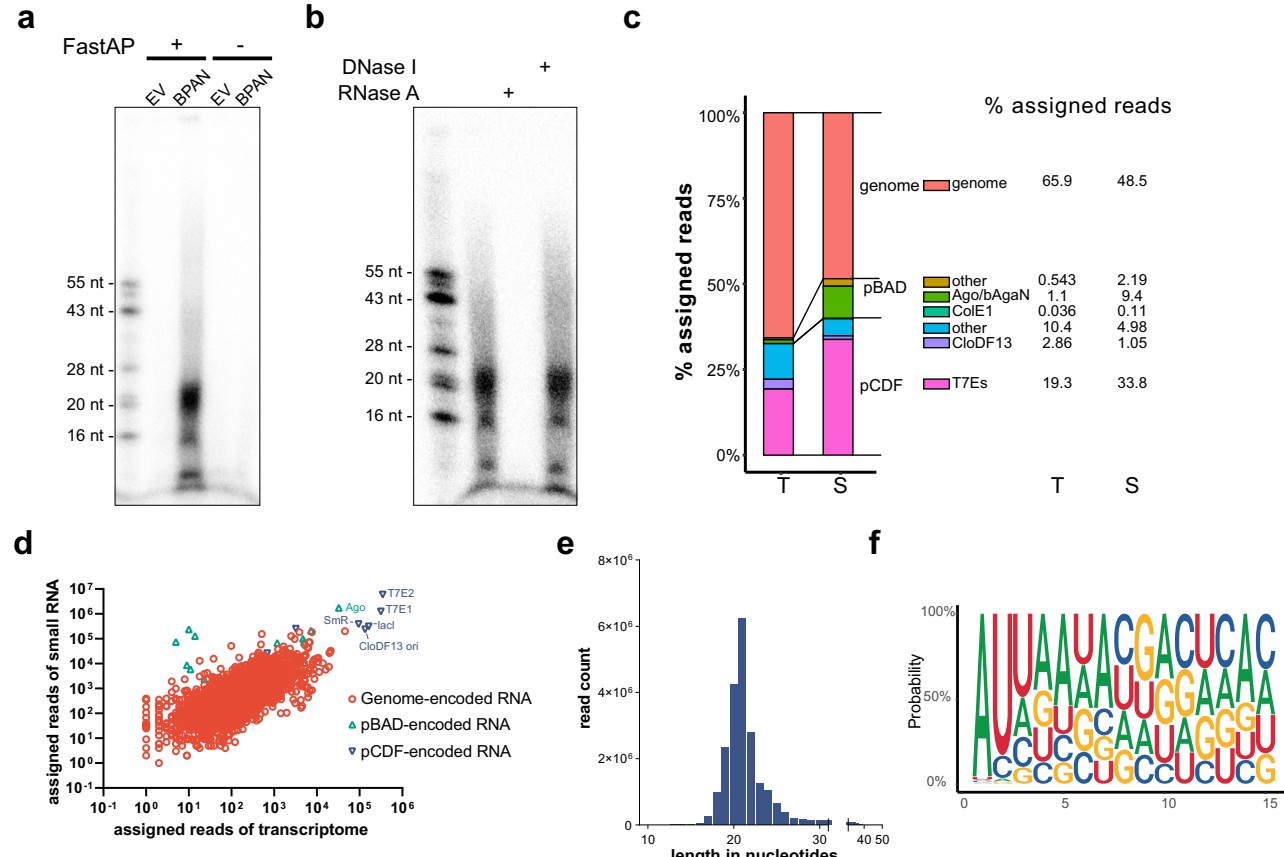

**Fig. 4 | EcAgo associates with small RNAs that are derived from transcriptome and contain 5'-end phosphate group. a** Analysis of the 5'-end of EcAgo-associated nucleic acids. Nucleic acids were extracted from the EcAgo protein sample that was purified from the cells containing pBAD24-EcAgo-EcbAgaN and pCDF-EGFP. The nucleic acids were treated with FastAP or not, labeled with γP32-ATP, and separated by denaturing polyacrylamide gel electrophoresis. **b** Treatment of the labeled nucleic acids with DNase I and RNase A, respectively. **c** Percentages of the transcriptome sequences (T) and small RNA sequences (S) that are assigned to the genome and specific plasmid elements. The elements are indicated with different colors, with the percentages shown on the right. **d** Correlation between the transcriptome sequences and small RNA sequences. The Pearson correlation coefficient is r > 0.79 with *p*-value < 10⁻⁹⁹. The *p*-value was calculated with two-sided *t*-test. **e** Length distributions of the small RNAs. **f** Nucleotide bias of the small RNAs. Source data are provided as a Source Data file.

To reveal whether EcbAgaN plays a role in the RNA acquisition of EcAgo, we analyzed the EcAgo-associated small RNAs from the cells containing pBAD24-EcAgo and pCDF-EGFP. The absence of EcbAgaN has no apparent effects on the length distributions or the sequence features of the small RNAs, or their correlation with the transcriptome, indicating that the RNA acquisition of EcAgo is independent of EcbAgaN (Supplementary Fig. 8). We also analyzed the EcAgo-associated small RNAs from the cells only containing pBAD24-EcAgo-EcbAgaN. The small RNAs show similar length distribution and sequence bias properties to those from the cells containing both the EcBPAN system and pCDF-EGFP, reinforcing that EcAgo does not specifically acquire small RNAs from the CloDF13 origin-derived transcripts (Supplementary Fig. 9).

**Guide-directed target recognition of EcAgo activates EcbAgaN**
To gain further insight into how the EcBPAN system senses invading genetic element and how it is activated, we performed an electrophoretic mobility shift assay (EMSA) to analyze the guide loading and target-binding properties of EcAgo. Single strand (ss) RNA and DNA substrates containing a 5' phosphate (5P) group and a 5' hydroxyl group (5OH) respectively were used in the assays. The results show that 5P-RNA is the preferred substrate of EcAgo (Fig. 5a), in agreement with that it associates with 5P-RNA in vivo. Moreover, when preloaded with 5P-RNA as a guide (gRNA), EcAgo can specifically recognize target ssDNA (TD), rather than nontarget ssDNA (NTD), target ssRNA (TR) or

nontarget ssRNA (NTR) (Fig. 5b). This suggests that although not detected in the EcAgo-copurified nucleic acids, ssDNA could serve the target of the EcBPAN system. We further analyzed whether EcAgo could mediate any DNA cleavage. Incubation of the ssDNA and dsDNA substrates with apo EcAgo or the EcAgo supplemented with guide RNA and/or target DNA does not result in any cleavage of the substrates (Supplementary Fig. 4e), indicating that EcAgo is not an active DNase.

The above data suggest that RNA-directed target DNA binding by EcAgo may activate the system. To reconstitute the activation in vitro, the EcAgo-gRNA (5P-RNA) complex (Supplementary Fig. 12a) was incubated with TD, NTD, TR or NTR. This resulted in the formation of EcAgo-gRNA-TD complex (Supplementary Fig. 12b). Then, the mixtures, as well as apo EcAgo and the EcAgo-gRNA complex, were analyzed for their effects on the DNase activity of EcbAgaN using genomic DNA as substrates (Fig. 5c). The results show that the DNase activity of EcbAgaN is significantly stimulated by the EcAgo-gRNA complex supplemented with TD instead of other oligonucleotides. The data indicate that target ssDNA recognition of guide-directed EcAgo activates EcbAgaN for nonspecific DNA degradation. In addition, the activated DNase activity was not affected by the presence or absence of ~200 mM NaCl or KCl, although the basal activity of EcbAgaN is inhibited by the salts of ~200 mM (Supplementary Fig. 12c).

To analyze whether EcAgo and EcbAgaN form a stable complex, we tried to pull down EcbAgaN using N-terminal tagged EcAgo from the cells co-expressing them. Nevertheless, EcbAgaN was not co-

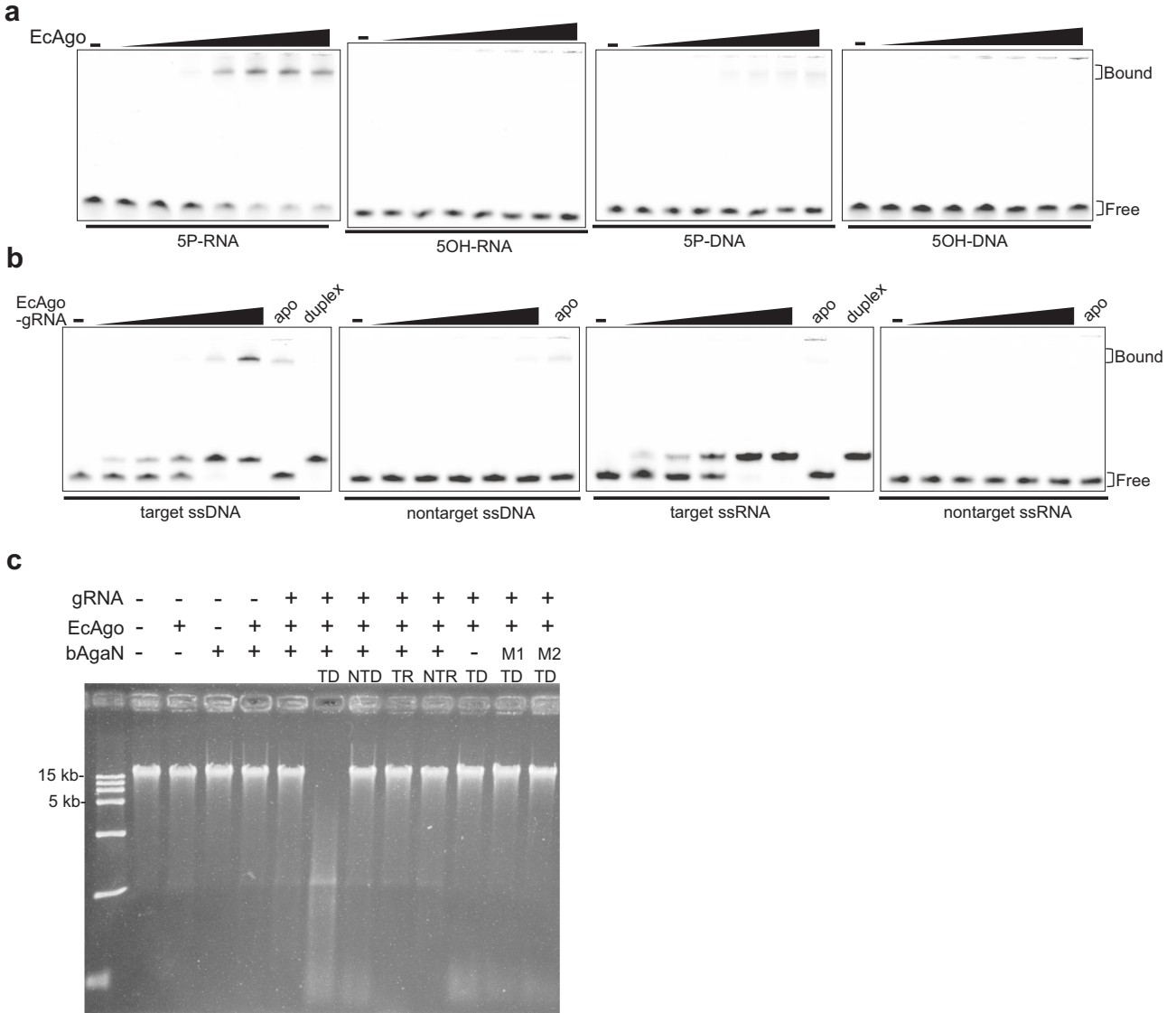

**Fig. 5 | EcAgo is directed by 5P-RNA guides to bind ssDNA targets and activates EcbAgaN. a** Nucleic-acid binding of EcAgo. The different substrates are indicated below the panels. **b** Target-binding of EcAgo preloaded with 5P-RNA as guide (gRNA). The different substrates are indicated below the panels. Binding of the substrates by apo EcAgo was also performed as controls. Duplex was pre-annealed with guide RNA and the corresponding target ssDNA or target ssRNA. **c** Target recognition of the gRNA-bound EcAgo activates EcbAgaN. About 300 ng genomic DNA was treated with EcbAgaN. Aliquots of the reaction were also supplemented with EcAgo, gRNA and/or target ssDNA or other oligonucleotides. TD: target ssDNA, NTD: nontarget ssDNA, TR: target ssRNA, NTR: nontarget ssRNA. The control reactions with EcAgo-gRNA-TD complex but lacking EcbAgaN or containing EcbAgaN mutants were also analyzed. Source data are provided as a Source Data file.

purified with EcAgo (Supplementary Fig. 13a), suggesting that EcAgo and EcbAgaN do not form a stable complex under experimental conditions.

## The GbBPAS system is activated by the CloDF13 origin to mediate NAD⁺ depletion

Next, we aimed to reveal whether BPAS systems also provide immunity against invading plasmid. bAgaS has been annotated as a Sir2_2-domain-containing protein and is only distantly related to the Sir2 domain of the short pAgo-associated Sir2-APAZ protein[7]. Structural prediction reveals that the Sir2_2 of domain GbbAgaS is similar to that of the ThsA protein of the Thoeris defense system[44] and they share the conserved residues of the NAD-binding site (Supplementary Figs. 2c, d, 3b). We constructed strains expressing individual GbAgo and GbAgaS proteins, GbBPAS system as well as the GbAgaS mutant (N155A) protein or system using the pBAD24

vector (Supplementary Fig. 6b). Alanine substitution of the corresponding Asparagine (N112) in ThsA can abolish its NAD⁺ degradation activity[44,45]. The abovementioned strains were further transformed with pCDF-EGFP to analyze the potential function of the plasmid. Then, the strains with or without pCDF-EGFP were plated onto the plates containing corresponding antibiotics and with or without the inducer (L-arabinose), and the cell viability was measured (Fig. 6a). The results show that individual GbAgaS induces a ~10⁴-reduction in the cell viability, which is independent of pCDF-EGFP but is abolished by GbAgo. On the other hand, the GbBPAS system only mediates cell death in the presence of pCDF-EGFP, indicating that pCDF-EGFP is the trigger. In addition, the N155A mutation abolishes the reduction of cell viability, indicating that the activity of the Sir2_2 domain is essential for the GbAgaS-mediated cell death. The variants of pCDF-EGFP were also tested for their ability to activate the GbBPAS system (Fig. 6b). This reveals that the

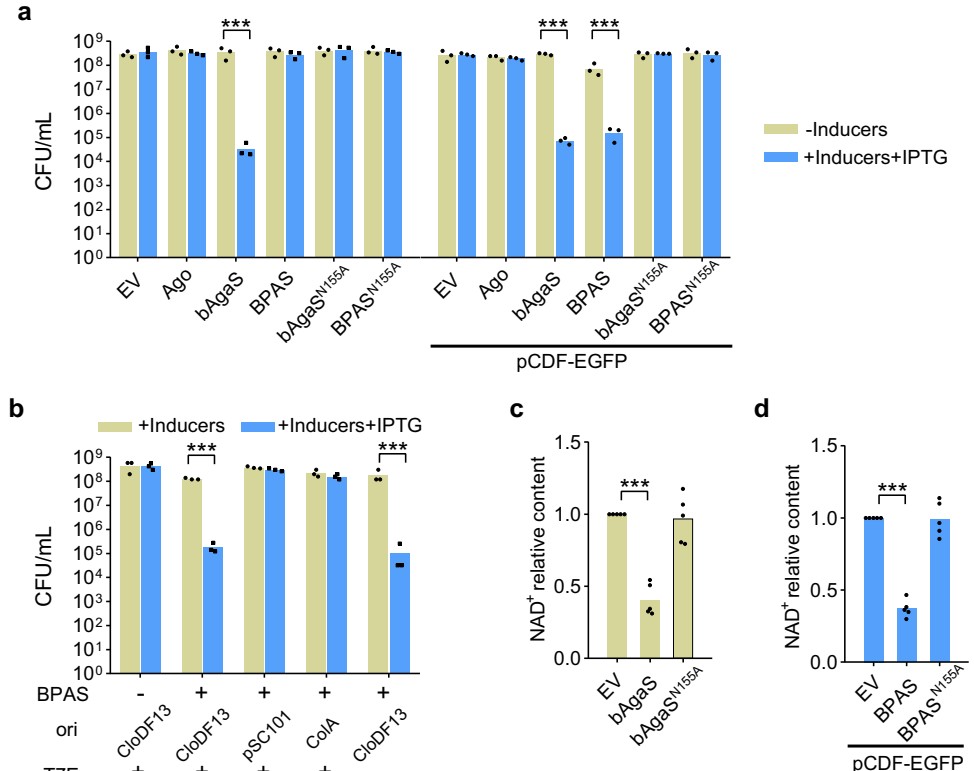

**Fig. 6 | GbBPAS system is activated by the CloDF13 origin and mediates NAD⁺ depletion. a** GbBPAS system is activated by pCDF-EGFP to induce cell death. The cells containing empty vector (EV), the wild type and mutant GbBPAS systems, and individual proteins as well as mutants in the presence or absence of pCDF-EGFP were plated onto the plates without inducer or with the inducer and IPTG. Then, CFU/mL was calculated. Data are presented as mean values with individual data points overlaid ($n = 3$ biological replicates). The p-values were calculated using one-sided Student's t-test [***: $p < 0.001$; $p = 0.00002444$, 8.778E-07, 0.0001522 (from left to right)]. Inducer: L-arabinose. **b** Activation of the GbBPAS system requires the CloDF13 origin. Cell viability of the cells carrying the GbBPAS system in the presence of pCDF-EGFP or its variants was measured. Data are presented as mean values with individual data points overlaid ($n = 3$ biological replicates). The p-values

were calculated using one-sided Student's t-test [***: $p < 0.001$; $p = 7.664E-06$, 0.0002523 (from left to right)]. **c** GbbAgaS reduces cellular NAD⁺ level. NAD⁺ amount of the cells carrying wild-type and mutant GbbAgaS proteins was measured, and relative NAD⁺ level was calculated with the values of the EV samples set as 1. Data are presented as mean values with individual data points overlaid ($n = 5$ biological replicates). The p-values were calculated using one-sided Student's t-test (***: $p < 0.001$; $p = 9.795E-07$). **d** GbBPAS system reduces cellular NAD⁺ level in the presence of pCDF-EGFP. NAD⁺ amount of the cells carrying wild-type and mutant GbBPAS systems was measured, and the relative NAD⁺ level was calculated. Data are presented as mean values with ($n = 5$ biological replicates). The p-values were calculated using one-sided Student's t-test (***: $p < 0.001$; $p = 6.549E-09$). Source data are provided as a Source Data file.

CloDF13 origin activates the GbBPAS system as observed for the EcBPAN system. Further, the pull-down assay reveals that GbAgaS was co-purified with N-terminal His-tagged GbAgo, indicating that the two proteins form a stable complex (Supplementary Fig. 13b).

Since many defense systems containing Sir2-like domains confer immunity by NAD⁺ depletion[22,23,44,45], we analyzed whether the GbBPAS system can also deplete NAD⁺. The cells expressing GbAgaS or its mutant, the GbBPAS system or the mutated system in the presence of pCDF-EGFP, were grown in liquid medium, and gene expression was induced by L-arabinose. The results show that GbAgaS individually and the GbBPAS system together with pCDF-EGFP induce significant culture growth retardation and NAD⁺ level reduction (Supplementary Fig. 14a, b, Fig. 6c, d). Moreover, the N155A mutation abolishes the reduction in NAD⁺ level and relieves the growth retardation. Together, the data indicate that the GbBPAS system confers immunity against invading plasmid via the Sir2_2 domain-mediated NAD⁺ depletion.

**The EaBPAM system mediates cell death depending on the transmembrane (TM) effector**

The functions of the EaBPAM system were analyzed using the same methods as applied for the GbBPAS system. Instead of a site mutation, we constructed a truncation mutant of EabAgaM (EabAgaM^ΔTM) that lacks the C-terminal TM region (106 a.a.) (Supplementary Figs. 2e, f,

6c). Pull-down assay did not detect physical interaction between EaAgo and EabAgaM (Supplementary Fig. 13c). Nevertheless, the EaBPAM system induces a ~10³-reduction in the cell viability, which requires both EaAgo and the full-length EabAgaM (Fig. 7a). The pCDF-EGFP plasmid is not required since the system expressed from the pBAD24 vector mediates cell death in the absence of pCDF-EGFP. Then, we expressed the system with pKD46, a low-copy vector, and analyzed its effect on cell viability. Again, the EaBPAM system induces significant cell death both in the presence and absence of other plasmids (Fig. 7b).

## Discussion

In this study, we characterized three types of long-B pAgo systems that comprise ~65% of all long-B pAgos (Fig. 1a). The systems are equipped with different associated effectors. Focusing on a long-B pAgo-nuclease (EcBPAN) system, we demonstrate that the system employs the nuclease to mediate genomic DNA degradation and trigger cell death upon recognition of invading nucleic acids, thus defending against the invader via Abi response. In the EcBPAN system, EcAgo can be directed by RNA guides to recognize ssDNA targets, which activates EcbAgaN for indiscriminate dsDNA degradation (Fig. 5). This efficiently kills the cells carrying the invader plasmid (Fig. 3d, reduction in cell viability on the antibiotic-free plates) and also depletes the invader plasmid from the cell population (Fig. 3d, additional reduction in cell viability on the

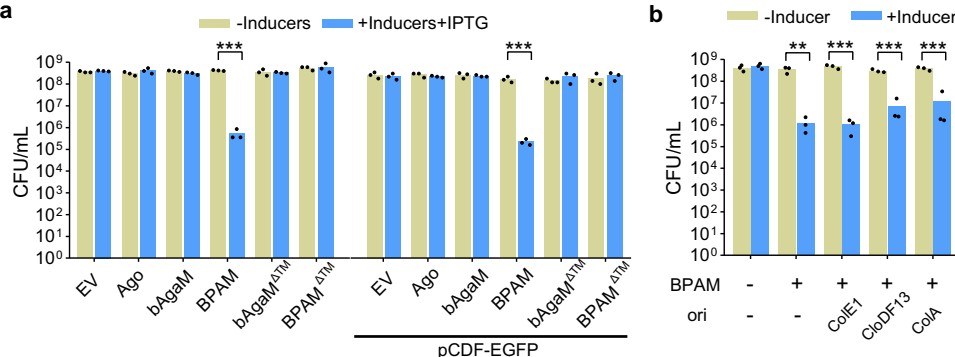

**Fig. 7 | EaBPAM system induces cell death. a** Cell viability of the cells containing empty vector (EV), the wild type and mutant EaBPAM systems, and individual proteins as well as mutants in the presence or absence of pCDF-EGFP was analyzed. Data are presented as mean values ($n = 3$ biological replicates). The $p$-values were calculated using one-sided Student's $t$-test [***: $p < 0.001$; $p = 0.00001019$, 6.380E-06 (from left to right)]. **b** Cell viability of the cells containing the EaBPAM system in the presence of pCDF-EGFP or its variants was measured. Data are presented as mean values ($n = 3$ biological replicates). The $p$-values were calculated using one-sided Student's $t$-test [**: $p < 0.01$; ***: $p < 0.001$; $p = 0.002786$, 0.0006713, 0.0003442, 0.0005144 (from left to right)]. Source data are provided as a Source Data file.

Str plates). The depletion of the invader may start with spontaneous plasmid loss; alternatively, the system may cleave the plasmid without killing the host in a few cells. Then, the invader-free cells have a selective advantage and finally outnumber the cells carrying the invader, such that the invader plasmid would be finally removed from the cell population. On the other hand, the BAC expression plasmid is retained in the survivors (Fig. 3d, no additional reduction in cell viability on the Chl plates), indicating that the EcBPAN system does not show biased cleavage activity against innocent extrachromosomal genetic elements.

Genomic DNA degradation is performed by some defense systems as an immune strategy[29–31], and potentially by more systems that employ nuclease effectors. In particular, a few type III CRISPR-Cas systems with NucC nucleases as effectors and type V-A2 CRISPR-Cas systems cleave genomic DNA after crRNA-directed target RNA recognition[29,30]. This suggests that a subgroup of pAgo and CRISPR-Cas, the two nucleic acid-directed defense systems, have convergently evolved to adopt the genomic DNA degradation Abi strategy. Moreover, the target-recognition-activated collateral DNA cleavage of the EcBPAN system has the potential to be repurposed for sequence-specific applications, as its CRISPR counterparts[29,46]. Further, the EcBPAN system is activated by a specific genetic element (the CloDF13 origin), indicating that its indiscriminate DNase activity can be manipulated in vivo, which will create opportunities to develop in vivo pAgo technologies. For example, the EcBPAN system has the potential to be repurposed as programmed cell death tools and random genome mutation or reduction tools.

Characterization of the EcAgo-associated nucleic acids sheds light on the mechanisms of how the EcBPAN system discriminates invaders from self. First, EcAgo acquires small 5P-RNA as guides, akin to the characterized long-B pAgo (RsAgo) and several short pAgo systems[22,24,27,28], while the guide acquisition does not require the associated nuclease. Second, the abundance of the guides is correlated with that of the RNAs in the transcriptome, but not biased toward the RNAs from the CloDF13 origin, the specific trigger element. The data indicate that EcAgo indiscriminately acquires guide RNAs from the transcriptome. Third, the CloDF13 origin and the ColE1 origin are of the same origin type that uses two RNAs to initiate plasmid replication and generates ssDNA region during the process[47,48]. However, only the former efficiently triggers the EcBPAN system to induce cell death. The phenomenon is possibly because activation of EcBPAN requires a high level of RNA guides, which can be produced from the CloDF13 origin but not the ColE1 origin. Finally, the requirement of the unique origin element

possibly reflects the generation of ssDNA targets during the replication initiation of the origin.

We additionally characterized GbBPAS system and EaBPAM system, which employ a Sir2_2-domain-containing protein and a transmembrane (TM) protein as effectors respectively. The GbBPAS system is activated by the CloDF13 origin, akin to the EcBPAN system, while the EaBPAM system, even when expressed from a low-copy plasmid pKD46 with a protein-primed replication mechanism[49], still induces cell death, indicative of a different invader discrimination mechanism. By comparison, the *Geobacter sulfurreducens* short pAgo system (GsSPARSA) is specifically activated by the CloDF13 origin, but its guides are enriched in the CloDF13 origin region by unknown mechanisms[22]. The *Maribacter polysiphoniae* short pAgo system (MapSPARTA) is activated by high transcription of multicopy plasmids[24]. Together, although all the characterized catalytically inactive pAgos are directed by 5P-RNA guides to recognize target ssDNA (Fig. 5)[22,24,27,28], the invader discrimination manners of these systems are diverse.

The GbBPAS system mediates NAD⁺ depletion in the presence of the invading plasmid via the Sir2_2 domain of GbbAgaS, in agreement with the canonical functions of Sir2-like domains in many defense systems[22–24,45]. GbbAgaS alone mediates dramatic cell death, which is suppressed by GbAgo possibly by direct interaction; then, GbAgo activates GbbAgaS after recognition of the invader plasmid. The regulation is in a similar manner to SPARTA systems[24]. By comparison, EcbAgaN or EabAgaM do not mediate cell death by themselves or form a stable complex with their respective pAgos under experimental conditions. The data suggest that the regulation of BPAN and BPAM systems may be different from BPAS systems.

In conclusion, we demonstrate that the BPAN system confers Abi immune response through the activation of the associated effector upon recognition of invading nucleic acids. Our data also suggest that BPAS and BPAM systems may employ a similar defense strategy using the Sir2 domain protein and the TM protein as their respective effectors. On the other hand, our phylogenetic analysis reveals that a minority of long-B pAgos cluster with other proteins, such as the og_54 VirE-N domain-containing protein, or do not have any associated protein in an operon (Fig. 1a). The biological functions and mechanisms of these long-B pAgos need to be addressed by future studies.

## Methods
### Bacterial strains and phages
*Escherichia coli* DH5a was routinely grown in Lysogeny broth (LB) medium and used for plasmid cloning, while *E. coli* BL21 (DE3) was

used for protein production and the in vivo assays. *E. coli* phages T5, T7 and Lambda-vir are gifts from Shi Chen lab (Wuhan University, China).

## Plasmid construction

In general, the plasmids were constructed with the restriction-ligation cloning method, assembled with ClonExpress II One Step Cloning Kit (Vazyme, Nanjing, China), or generated via the site-directed mutagenesis method. For restriction-ligation cloning, the gene fragments were amplified using the primers containing restriction sites (Supplementary Data 3), digested by restriction enzymes and inserted into the plasmid vectors between the corresponding sites. For ClonExpress assembly, the fragments of gene coding sequences, replication origins, and expression cassettes were amplified with the primers that have homologous sequences with the backbone vectors, and the linear vectors were prepared either by restriction digestion or by PCR. Then, the fragments were assembled following the instructions of the manufacturer. For site-directed mutagenesis, the plasmid fragments were generated via PCR using the primers containing mutant sites (Supplementary Data 3) and transformed into *E. coli* DH5α after the template plasmids were removed by DpnI.

The plasmids listed in Supplementary Data 2 were used as vectors to express long-B pAgos and their associated proteins and to construct the trigger plasmids. The primers used for plasmid construction were listed in Supplementary Data 3 with the cloning methods indicated. All cloning work was performed with *E. coli* DH5α. All the primers were synthesized by Tingke (Beijing, China).

Specifically, the optimized coding sequences of long-B pAgos and their associated proteins were synthesized by Tingke (Beijing, China). To obtain EcAgo and EcbAgaN proteins for biochemical analysis, the coding sequences of EcAgo were inserted into pCDFDuet-1, while the coding sequence of EcbAgaN was inserted into pET28a, allowing the proteins to be expressed from an IPTG-induced promoter with N-terminal 6×His tag. Site mutagenesis was performed by overlapping PCR.

For in vivo experiments, pET28T, where the T7 promoter and lacO operator were replaced by a TetR expression cassette and a Tet promoter from the p46Cpf1-OP2 plasmid (Addgene #98592), was used to express EcbAgaN and its mutants, as well as His-tag-free EcbAgaN (pET28T-EcbAgaN_HF). In addition, the coding sequences of EcAgo were also inserted into pBAD24 to express the proteins under the control of an araBAD promoter. To express the EcBPAN system using pBAD24, a fragment containing TetR expression cassette, Tet promoter, EcbAgaN coding sequence and the terminator was amplified and inserted into pBAD-EcAgo. The plasmids expressing mutant EcBPAN systems were constructed in a similar way.

We also constructed pBAD24 expression plasmids that express both EcAgo and EcbAgaN using araBAD promoter (pBad24-EcAgo-aEcbAgaN). To this end, a fragment containing the araBAD promoter and *EcbAgaN* gene was generated by overlapping PCR and then inserted into pBAD-EcAgo by restriction-ligation cloning. The resulting plasmid was used as template to amplify the EcBPAN expression cassette, which was then inserted into pBeloBAC11 by ClonExpress assembly to generate pBAC-EcAgo-EcbAgaN. The pBAC expression plasmids carrying the mutated EcBPAN systems were constructed in a similar way.

To generate the pCDF-ΔT7 plasmid, the DNA fragment lacking T7 expression boxes was amplified from pCDF-EGFP. Then, recircularization of the fragment was performed by ClonExpress II One Step Cloning Kit. To replace the origin of pCDF-EGFP plasmids with ColA or pSC101^{TR}, the DNA fragments containing ColA or pSC101^{TR} replicons, and the backbone of pCDF-EGFP plasmid were amplified respectively, and assembled using the ClonExpress kit.

For the in vivo experiments of the GbBPAS system and EaBPAM system, the coding sequences of long-B pAgos and their associated proteins were inserted into pBAD24 respectively. Then, the expression cassettes of GbbAgaS and EabAgaM were amplified and inserted into pBAD24-GbAgo and pBAD24-EaAgo, respectively. The site mutagenesis of GbbAgaS was performed by overlapping PCR, while the truncation of the *EabAgaM* gene was performed by amplification of the fragment lacking the gene C-terminal region from pBAD24-EaAgo-EabAgaM and pBAD24-EabAgaM respectively and recircularization of the fragment.

To construct the pKD46-M1-EaAgo-EabAgaM plasmid, the p^{TR}Cas12a-NT plasmid from our previous study[50] was used as template to amplify the Chl-resistant gene and the araBAD promoter-araC-pSC101 origin fragment. Then, the fragments were assembled with the fragment containing *EaAgo* and *EabAgaM* genes amplified from pBAD24-EaAgo-EabAgaM.

To construct the plasmids expressing His-tagged GbbAgaS/EabAgaM, their coding sequences were amplified and inserted into pET21a, generating pET21a-GbbAgaS and pET21a-EabAgaM. The plasmids expressing mutant GbbAgaS and EabAgaM were constructed by the site-directed mutagenesis method. Specifically, pET21a-GbbAgaS and pET21a-EabAgaM were used as template to amplify the plasmid fragments containing the mutated sites. After the templates were digested by DpnI, the plasmid fragments were transformed into *E. coli* DH5α, resulting pET21a-GbbAgaS_N155A and pET21a-EabAgaM_ΔTM, respectively.

To construct the co-expression plasmids for the pull-down assay, the coding sequences of the His-tag of pET28a-EcbAgaN, pET21a-GbbAgaS and pET21a-EabAgaM were replaced by the coding sequences of HA-tag, through the site-directed mutagenesis method. Then, the N-terminal His-tagged Ago expression cassettes were amplified and inserted in the abovementioned pET28a/ pET21a plasmids via restriction-ligation cloning.

## Protein expression and purification

The plasmids used for the purification of EcAgo and EcbAgaN proteins and their variants were transformed into *E. coli* BL21(DE3). The transformants were grown in LB medium at 37 °C containing the corresponding antibiotics. At an optical density (OD_{600}) of ~1.0, the cultures were cooled on ice for 10 min and then gene expression was induced with 0.4 mM IPTG at 16 °C for 18 h. The purification procedure was modified from the published protocol of our lab, and the common steps include cell extract preparation, Ni-NTA affinity chromatography (NAC), anion exchange chromatography (AEC) and size exclusion chromatography (SEC)[21]. To prepare cell extract, the cell mass was resuspended in 50 mL of lysis buffer (20 mM HEPES pH 7.5, 20 mM imidazole, 500 mM NaCl) and disrupted by French press, followed by centrifugation at 13,000×*g* for 40 min to remove cell debris. NAC was performed using Ni-NTA agarose resin columns (Cytiva, Marlborough, MA, USA), from which His-tagged proteins were eluted using a lysis buffer containing 300 mM imidazole. A 5 mL Q FF column (Cytiva, Marlborough, MA, USA) was used for AEC and the proteins were eluted using a 35 mL linear gradient of 0–1 M NaCl. For SEC, the AEC fractions containing target proteins were collected and loaded onto a Superdex 200 column (Cytiva, Marlborough, MA, USA). Finally, the proteins were eluted with Buffer C (20 mM Tris-HCl pH 7.5, 250 mM NaCl) and analyzed by SDS-PAGE.

In addition, an ammonium sulfate precipitation step (ASP) was performed before NAC to remove the nucleic acids in EcAgo. Specifically, the cell extract was slowly supplemented with saturated ammonium sulfate solution up to the final saturation of 55%. Then, the cell extract was incubated on ice for 1 h and the proteins were collected by centrifugation at 13,000×*g* for 40 min. The proteins were resuspended by 10 mL of lysis buffer and subjected to NAC and the following purification steps as described previously[21]. Hereafter, all pH values were adjusted at 25 °C.

## Western blot analysis of the Ago-associated proteins and their mutants

The plasmids expressing His-tagged EcbAgaN, GbbAgaS, EabAgaM and their variants were transformed into *E.coli* BL21(DE3). Single colonies were cultured in 10 mL LB containing the corresponding antibiotic to OD600 of 0.6 at 37 °C. Then, gene expression was induced with 0.5 mM IPTG at 37 °C for 3 h. The cells were collected by centrifugation and then resuspended in 200 μL of lysis buffer. The cell solution was subjected to sonication in a cold bath (amp 20%, 2 s ON/2 s OFF, 2 min duration). After centrifugation at 13,000×*g* for 5 min, 5 μL of the supernatant from each sample was mixed with 5X SDS-loading dye and boiled at 95 °C for 10 min. The proteins were separated by 12% SDS-PAGE and transferred onto a PVDF membrane (Bio-Rad, Hercules, CA, USA). The membrane was blocked in TBST buffer (50 mM Tris-HCl, 100 mM NaCl, 0.05% Tween 40, pH 8.0) containing 5% skimmed milk, and then incubated with anti-His antibody (catalog number: ABT2051, Abbkine, Wuhan, China) at a 1:5000 dilution in TBST containing 5% skimmed milk. Upon completion of three consecutive washing steps, the membrane was incubated with the secondary antibody (Goat Anti-Rabbit IgG, catalog number: A21020, 1:5000 dilution) (Abbkine). After removing unspecific binding, the signal was detected with the Clarity Western ECL Substrate (Bio-Rad, Hercules, CA, USA) and Monad QuickChemi 5200 chemiluminescence imaging system (Monad Biotech, Shanghai, China).

## Pull-down assay (co-purification)

The plasmids expressing His-tagged Ago, or the HA-tagged associated protein or both of them were transformed in *E.coli* BL21(DE3). Gene expression was induced with 0.5 mM IPTG at 37 °C for 3 h. Cell extract was prepared and the His-tagged proteins were purified with NAC as described previously[21]. Then, the cell extracts and the eluates of NAC were analyzed by SDS-PAGE and western blot. Western blot was performed as described in the "Western blot analysis" section, except that anti-His antibody was replaced with primary rabbit anti-HA antibody (catalog number: ABT2041, Abbkine, Wuhan, China).

## Size exclusion chromatography–multi-angle light scattering

SEC-MALS (Size Exclusion Chromatography with Multi-Angle Light Scattering) was used to analyze the oligomer size of EcbAgaN. Specifically, after being purified with SEC, EcbAgaN was loaded onto a Superdex 200 10/300 GL column (Cytiva, Marlborough, MA, USA) pre-equilibrated with 20 mM Tris-HCl pH 7.5, 250 mM NaCl, at 0.5 mL/min flow rate. Then, Wyatt Dawn Heleos II detector (Wyatt Technology, Santa Barbara, CA, USA) collected the static light scattering, while the absorbance at 280 nm was monitored by AKTA pure 25 UV detector (Cytiva, Marlborough, MA, USA). Data were collected and analyzed in ASTRA 7 software (Wyatt Technology, Santa Barbara, CA, USA). BSA monomer is used as a known molecular weight standard.

## Nuclease assay

The oligonucleotides used as substrates or to prepare the substrates in the nuclease assay are listed in Supplementary Data 4. The substrate's genomic DNA and pUC19 plasmid were extracted from *E. coli* DH5a. To analyze the substrate specificity of EcbAgaN, 100 nM FAM-labeled single strand (ss) DNA (OXS1), double strand (ds) DNA (OXS1/OXS2), ssRNA(OXS3), dsRNA (OXS3/OXS4) and DNA/RNA (OXS5/OXS4 or OXS3/OXS6) duplexes were incubated with 200 nM EcbAgaN in the presence of 20 mM Tris-HCl (pH 7.5), 25 mM NaCl, 5 mM MgCl$_2$ and 5 mM MnCl$_2$ at 37 °C for 10 min. Then, The reactions were treated with 2 mg/mL Proteinase K (Thermo Fisher Scientific, Waltham, MA, USA) in the presence of 5 mM CaCl$_2$ at 55 °C for 30 min and were analyzed by denaturing polyacrylamide gel electrophoresis, and the gel was imaged by a Typhoon 5 laser-scanner (Cytiva, Marlborough, MA, USA). The same reaction mixtures containing dsDNA (OXS1/OXS2) and

ssDNA (OXS1) as substrates were also used to analyze the activity of EcbAgaN mutants. In the assays, the monomer concentration of EcbAgaN was calculated.

To analyze the metal dependency of EcbAgaN, 100 nM FAM-labeled dsDNA (OXS1/OXS2) was incubated with 200 nM EcbAgaN in the presence of 20 mM Tris-HCl (pH 7.5), 25 mM NaCl, as well as indicated metal ions or EDTA at 37 °C for 10 min. The reactions were analyzed as described above.

In the time course assay, 100 nM 5'FAM-labeled ssDNA (OXS1), 3'FAM-labeled ssDNA (OXS18), 5'FAM-labeled dsDNA (OXS1/OXS2) and 3'FAM-labeled dsDNA (OXS2/OXS18) were incubated with 50 nM EcbAgaN in the presence of 20 mM Tris-HCl (pH 7.5), 25 mM NaCl, and 5 mM MnCl$_2$ at 37 °C for 1 min, 2 min, 5 min, 10 min, 20 min and 40 min, respectively. The reactions were analyzed as described above.

To analyze whether EcbAgaN degrades plasmid and genomic DNA, ~150 ng pUC19 (~9 nM in the final reaction mixture) and 300 ng *E. coli* genomic DNA were incubated with a gradient of EcbAgaN (0, 50, 100, 200 nM) at 37 °C for 40 min, respectively. Then, the samples were treated with 2 mg/mL Proteinase K and separated by agarose gel electrophoresis. The gels were stained with EtBr. and imaged using the Molecular Imager Gel Doc EX system (NewBio Industry, Tianjin, China). The pUC19 cut by restriction enzyme *Eco*R I and nicking endonuclease Nt.*Bsp*Q I was used as linear (L) DNA and open circular (OC) DNA controls respectively.

To analyze the effects of salt concentrations on the nuclease activity of EcbAgaN, ~300 ng genomic DNA and 50 nM EcbAgaN were incubated at 37 °C for 40 min in the presence of 20 mM Tris-HCl (pH 7.5), 25 mM NaCl and 5 mM MnCl$_2$ with a gradient additional NaCl and KCl (0, 50, 100, 150, 200, 300 mM) respectively. Then, the samples were treated with 2 mg/mL Proteinase K and analyzed by agarose gel electrophoresis.

To analyze the possible nuclease activity of EcAgo, 100 nM labeled ssDNA (OXS1) and dsDNA (OXS1/OXS2) substrates were incubated with 500 nM EcAgo, with or without 100 nM unlabeled 5P-RNA (OXS11) and target DNA (OXS14) at 37 °C for 1 h. Then, the reactions were treated with 2 mg/mL Proteinase K (Thermo Fisher Scientific, Waltham, MA, USA) in the presence of 5 mM CaCl$_2$ at 55 °C for 30 min. At last, the samples were analyzed by denaturing polyacrylamide gel electrophoresis as described above.

To analyze whether EcbAgaN is activated by EcAgo upon guide and targeting binding, 500 nM EcAgo was first incubated with 500 nM 5P-RNA (OXS11) guide at 37 °C for 15 min, in the presence of 20 mM Tris-HCl (pH 7.5), 225 mM NaCl, 5 mM MgCl$_2$ and 5 mM MnCl$_2$ followed by a subsequent incubation with 500 nM target ssDNA (OXS14), nontarget ssDNA (OXS15), target ssRNA (OXS16) and nontarget ssRNA (OXS17) at 37 °C for 15 min respectively. Then, ~300 ng genomic DNA and 50 nM EcbAgaN were supplemented into the reaction mixtures, which were then incubated at 37 °C for 1 h. The incubation with the protein storage buffer, water or dead EcbAgaN was performed as controls. At last, the samples were treated with 2 mg/mL Proteinase K and analyzed by agarose gel electrophoresis as described above. To analyze whether EcAgo indeed forms complexes with 5P-RNA or with 5P-RNA and target ssDNA in the reaction mixtures, FAM-labeled 5P-RNA and FAM-labeled target ssDNA were used to replace their non-labeled counterparts and incubated with EcAgo as described above. Then, the reaction mixtures were analyzed as described in the "Electrophoretic mobility shift assay" section before EcbAgaN and genomic DNA were supplemented.

To analyze the effects of salt concentrations on the activation of EcbAgaN, ~300 ng genomic DNA was degraded by 50 nM EcbAgaN with or without pre-incubated EcAgo-OXS11-OXS14 in the presence of the indicated salt concentrations (Supplementary Fig. 12c). Then, the samples were treated with 2 mg/mL Proteinase K and analyzed by agarose gel electrophoresis.

## Electrophoretic mobility shift assay (EMSA)

To analyze the nucleic acid binding properties of EcAgo, 100 nM of four different 3'-FAM-labeled nucleic acid substrates (5P-DNA (OXS8), 5OH-DNA (OXS7), 5P-RNA (OXS10) and 5OH-RNA (OXS9)) were incubated with a gradient of EcAgo (25, 50, 100, 200, 400, 800, 1600 nM), respectively, in a 10 µL mixture containing 20 mM Tris-HCl pH 7.5, 5 mM MgCl$_2$, 225 mM NaCl and 1 mM DTT at 37 °C for 30 min. After incubation, the reaction samples were supplemented with 2 µL loading dye containing 50% glycerol, 0.1% bromophenol blue and 0.1% xylene cyanol, and loaded onto 8% native polyacrylamide gels. The electrophoresis was performed in Tris–glycine buffer (25 mM Tris, 192 mM glycine) with 5 mM MgCl$_2$ at 100 V for 1 h. At last, the fluorescent signal was visualized using Amersham Typhoon 5 (Cytiva, Marlborough, MA, USA).

To analyze the target-binding specificity of the EcAgo-guide complex, 800 nM EcAgo was incubated with 100 nM unlabeled 5P-RNA (OXS11) at 37 °C for 15 min. Then, aliquots of the mixture were diluted to 50 nM, 100 nM, 200 nM, and 400 nM EcAgo, respectively. After that, the mixtures were incubated with 100 nM of FAM-labeled target DNA (OXS5), nontarget DNA (OXS12), target RNA (OXS3) and nontarget RNA (OXS13) at 37 °C for 40 min, respectively. The DNA/RNA duplex and dsRNA were generated from FAM-labeled target DNA (OXS5) and target RNA (OXS3) with their complementary pairing unlabeled 5P-RNA (OXS11), respectively, and run in the gel as controls. The reactions were also analyzed by native polyacrylamide gel electrophoresis as described above.

## Cell viability assay

In general, the effects of the long-B pAgo systems on cell viability were analyzed in two ways, i.e., gene expression was induced when the cells were grown on plates or in liquid medium, hereafter, the plate induction assay and the liquid medium induction assay respectively.

For the plate induction assay, single colonies of the transformants containing individual long-B pAgos and the associated proteins, the long-B pAgo systems and/or the trigger plasmids were grown in LB medium containing corresponding antibiotics at 37 °C overnight. The cultures (0.1 mL) were transformed into fresh medium (10 mL) containing corresponding antibiotics and grown at 37 °C for ~3 h. Then, the cells were serially diluted and dropped onto LB agar plates containing the corresponding antibiotics and inducers. After the plates were incubated at 37 °C for 16 h, the numbers of the colonies were counted and colony formation units per mL (CFU/mL) were calculated. The data of the plate induction assays are shown in Figs. 3e, 6a, b, 7 and Supplementary Fig. 7c.

For the liquid medium induction assay, single colonies of the transformants containing individual long-B pAgos and the associated proteins, the long-B pAgo systems and/or the trigger plasmids were grown in LB liquid medium containing corresponding antibiotics overnight. The cultures were transformed into fresh medium at a ratio of 1:100 with corresponding antibiotics (Fig. 6c, d, Supplementary Fig. 14) or without them (Fig. 3a–d, Supplementary Fig. 7a, b). The cultures were grown for ~60 min up to an OD$_{600}$ of ~0.1, when the inducers, including L-arabinose, aTc and/or IPTG as indicated in the figure legends, were supplemented. Then, aliquots of the cultures were sampled at indicated time points to analyze cell viability and plasmid maintenance, genomic DNA integrity and NAD$^+$ level as indicated in figure legends. Cell viability and plasmid maintenance were analyzed by measuring CFU/mL on antibiotic-free plates and the plates containing corresponding antibiotics, respectively.

## Genomic DNA extraction and analysis

The cells expressing wild-type EcBPAN system with pCDF-EcAgo and pET28T-EcbAgaN, or the mutant systems using pET28T-EcbAgaN-M1 and pET28T-EcbAgaN-M2 instead, or containing the corresponding empty vectors were transformed from overnight cultures into 100 mL

antibiotic-free LB medium. At an OD$_{600}$ of ~0.1, gene expression was induced by 80 ng/mL aTc and 50 µM IPTG. At 0, 0.5, 1, 2 h post-induction (hpi), cells from 2 mL of the cultures were collected by centrifugation. Genomic DNA was extracted using HiPure Bacterial DNA Kit (Magen, Guangzhou, China) following the manufacturer's instruction, and at the final step, the genomic DNA was eluted with 30 µL water. Then, 1 µL of the DNA samples were loaded on a 1% agarose gel, run for 40 min at 100 V in 1x TBE buffer and stained with EtBr. The results were imaged and analyzed using the Molecular Imager Gel Doc EX system (NewBio Industry, Tianjin, China). The genomic DNA from the cells expressing EcBPAN using BAC plasmid was analyzed in the same way, except that the inducer was 0.2% L-arabinose.

## Flow cytometry analysis of DNA content distributions

The cultures were prepared as described in the "Genomic DNA extraction and analysis" section, and the samples for flow cytometry analysis were prepared following the protocol established in our group with some modifications[21]. Specifically, 300 µL of the cultures were mixed with 700 µL absolute ethanol at 2 hpi and incubated at 4 °C overnight. Before being stained, the cells were collected by centrifuging at 2500×g for 5 min and washed with 1 mL 1x PBS buffer. The cells were collected by centrifugation again and resuspended in 30 µL 1x PBS buffer supplemented with 2 mg/mL DAPI (Thermo Scientific, Waltham, MA, USA), and stained for at least 1 h on ice in darkness. Then, the cell suspensions were diluted to a final volume of 1 mL by 1x PBS buffer and loaded onto a cytoflex-LX flow cytometer (Beckman Coulter, Brea, CA, USA) with a 375 nm laser, and a dataset of at least 20,000 cells was recorded for each sample. For each cell, the values of fluorescence signal at 450 nm (DAPI signal), FSC (forward scattered light), and SSC (side scattered light) were measured. The results were analyzed and visualized by FlowJo v.10.8.1 (BD Biosciences, Franklin Lakes, NJ, USA).

## Quantification of cellular NAD+ level

The cells expressing EabAgaS or its mutant, or EaBPAS system or the mutant system in the presence of pCDF-EGFP, or containing pBAD24 as empty vector were grown in corresponding antibiotics as described in the liquid medium induction assay. At an OD$_{600}$ of ~0.25, gene expression was induced by 0.2% L-arabinose. At 2 hpi, the OD$_{600}$ of cultures were normalized to ~0.4, and the cells from 1 mL of the cultures were collected by centrifugation and washed with 1 mL 1x PBS buffer. The cells were subjected to the measurements of NAD$^+$ level using the Coenzyme I NAD(H) Content Assay Kit (Solarbio, Beijing, China) following the instructions of the manufacturer. Specifically, the cells were resuspended by 500 µL of the acid extraction buffer (Solarbio, Beijing, China), and lysed by the sonicator bath (amp 20%, 2 s ON/2 s OFF, 2 min duration) at 4 °C. The cell extracts were centrifuged to remove cell debris at 12,000×g for 10 min, and the NAD$^+$ level of supernatant was quantified by the MTT (Methyl Thiazolyl Tetrazolium) assay following the instructions. Finally, the OD at 570 nm of each sample (OD$^{sample}$) and the corresponding control (OD$^{sample\_control}$), where the NAD$^+$ in the sample was neutralized before supplementation of MTT, was measured. The relative NAD$^+$ content of each sample was calculated using the Eq. 1:

$$\text{Relative NAD}^+ \text{ content} = \left(\text{OD}^{sample} - \text{OD}^{sample\_control}\right) \Big/ \left(\text{OD}^{EV} - \text{OD}^{EV\_control}\right)$$
(1)

Five biological replicates were performed for the NAD$^+$ level assay. EV: empty vector.

## Purification of EcAgo for nucleic acid extraction

The cells containing pBAD24 + pCDF-EGFP (EV), pBAD24-EcAgo + pCDF-EGFP, pBAD24-EcAgo-EcbAgaN, pBAD24-EcAgo-EcbAgaN + pCDF-EGFP were grown in LB medium containing corresponding

antibiotics at 37 °C up to an OD$_{600}$ of ~0.8. Protein expression was induced by 0.2% L-arabinose for EcAgo and 80 ng/mL aTc for EcbAgaN at 37 °C for 2 h. For the cells containing pCDF-EGFP, 200 μM IPTG was also supplemented to induce the T7 promoter. The cells were collected by centrifugation, resuspended by Buffer A (20 mM HEPES-NaOH pH 7.5, 5 mM MgCl$_2$, 5 mM MnCl$_2$, 100 mM NaCl and 5% glycerol) and disrupted by French press. The cell extracts were applied to 1 mL HiTrap TALON crude column (Cytiva, Marlborough, MA, USA). The column was washed with ~20 mL Buffer A containing 5 mM imidazole, followed by ~20 mL Buffer A containing a linear gradient of imidazole from 10 to 50 mM. Then, the proteins were eluted in Buffer A with 300 mM imidazole. Meanwhile, the cells from 10 mL of the same culture were also collected and used for the extraction of total RNA, respectively.

### Extraction of EcAgo-copurified nucleic acids
To extract the copurified nucleic acids, 500 μL of the protein solution was treated with 200 μg/mL Proteinase K for 1 h and then supplemented with 500 μL phenol/chloroform/isoamyl alcohol (pH 8.0, 25:24:1), followed by a brief vortexing. The sample was centrifuged at 16,000×$g$ for 20 min at 4 °C. The upper phase (about 400 μL) was transferred into a new tube and mixed with 40 μL 3 M NaAc (pH 5.2) and 500 μL isopropanol. After incubating at −20 °C for 1 h, The sample was centrifuged at 16,000×$g$ for 20 min at 4 °C. The pellet was washed with 1 mL pre-cooled 70% ethanol and dried for 30 min at room temperature. Finally, the nucleic acids in the pellets were resuspended in 50 μL DEPC water.

### Labeling and treatment of the EcAgo-copurified nucleic acids
One μL of the nucleic acids was treated with FastAP Thermosensitive Alkaline Phosphatase (Thermo Fisher Scientific, Waltham, MA, USA) in a 10-μL mixture containing 1 μL of 10X Buffer (Thermo Fisher Scientific) at 37 °C for 30 min. Mock treatment using water instead of FastAP was performed as controls (- FastAP). Then, the samples were heated at 90 °C for 10 min. One μL of the heated samples was labeled with γ$^{32}$P-ATP (PerkinElmer, Waltham, MA, United States) by T4 polynucleotide kinase (PNK, Thermo Scientific) using the forward reaction buffer at 37 °C for 20 min. Then, the samples were analyzed by denaturing polyacrylamide gel electrophoresis. The gel was exposed to a phosphor screen and imaged by a Typhoon 5 laser-scanner (Cytiva, Marlborough, MA, USA).

The labeled nucleic acids were further treated with RNase A (DNase and protease-free, Thermo Scientific, EN0531) or DNase I (RNase-free, Thermo Scientific, EN0521) for 1 h at 37 °C, and analyzed by denaturing polyacrylamide gel electrophoresis and autoradiography as described above.

### RNA sequencing and analysis
Library construction and sequencing were performed by Novogene Bioinformatics Technology Co., Ltd (Beijing, China). Specifically, rRNA was removed by Illumina Ribo-Zero Plus rRNA Depletion Kit (NEB, USA), and the libraries for transcriptome analysis were generated with NEBNext Ultra RNA Library Prep Kit for Illumina (NEB, USA). The library quality was assessed on the Agilent 5400 system (Agilent, USA). The qualified libraries were sequenced by Illumina NovaSeq6000 sequencing with PE150 strategy (paired-end reads and 150 bp read length).

Small RNA sequencing libraries were generated using NEBNext® Multiplex Small RNALibrary Prep Set for Illumina® (Set 1) (NEB, USA). After ligased with adapters, the RNAs were converted into DNA by reverse transcription following the protocol of the manufacturer, which was enriched by 14 cycles of PCR. Analysis of the PCR products did not detect larger molecular weight products (>500 bp), and the PCR products between 130 and 160 bp were pooled as the final library.

The final library was sequenced by Illumina NovaSeq6000 sequencing with SE50 strategy (single-end reads and 50 bp read length).

We used Fastp (version 0.23.1)[51] to process the raw reads with default parameters, including trimming the adapter sequences. The processed paired-end reads of the transcriptome sequencing were aligned to the genome of *E. coli* BL21 (GenBank: CP053602.1) and to the expression plasmids (pCDF-EGFP and/or pBAD24-EcAgo or pBAD24-EcAgo-EcbAgaN) using HISAT2 v2.1.0 (default parameters)[52]. The single-end reads of the small RNA sequencing were aligned to the above genomes and expression plasmids using bowtie v1.3.1[53], with the -v parameter limiting the number of mismatched bases to 1 and other parameters as default. The alignment generated bam files that were processed by samtools stats[54] to analyze the length distributions of small RNAs. To analyze the nucleotide frequency distributions, the bam files were converted into fasta files that were processed by seqkit[55] to intercept the 1-15 nt from the mapped reads with the length ≥15 nt. Then, the nucleotide frequency distributions of the intercepted reads were visualized using the R package 'ggseqlogo'. The analysis of the alignments to the origins and expression cassettes was performed in the same way. FeatureCounts[56] was used to assign reads to genomic features (default parameters). The assignment using featureCounts may lead to biased results upon multicopy genes. For example, the reads of the *lacI* gene from the pCDF plasmid can be assigned to the *lacI* gene of the genome. The code used to prepare the figures has been uploaded to https://github.com/yanqiuLiu0908/BPAN-system-analysis.git.

### Phylogenetic analysis
**Gene neighborhood (operons) analysis and protein sequences.** The accession numbers of all the long-B pAgos from the previous study[7], 199 proteins in total, were used as queries to find their sequences in NCBI on October 1, 2022. However, only 192 items were found and their sequences were downloaded from Genbank using Batch Entrez (https://www.ncbi.nlm.nih.gov/sites/batchentrez?). To analyze the associated proteins of long-B pAgos, the neighborhood containing 10 genes from both upstream and downstream of long-B *pAgo* genes was analyzed manually. This reveals that the og_15, og_44 and og_100 genes are invariably organized in the same operon with long-B *pAgos* with conserved operon structure (Fig. 1e), while *og_54* is usually located between the 2nd and 8th genes upstream of long-B *pAgos*, as exemplified by the two genes clusters in Supplementary Fig. 1. Then if the upstream or downstream genes do not encode any protein from the four ogs, we analyzed whether the long-B *pAgos* form operon-like structures with their adjacent genes and whether the adjacent genes encode proteins belonging to other ogs. If so, the long-B pAgos are marked as other in Fig. 1a (representative gene clusters shown in Supplementary Fig. 1). If not, the long-B pAgos are considered to have no associated proteins (marked as none in Fig. 1a, Supplementary Fig. 1). The protein sequences of og_15, og_44 and og_100 members were also downloaded from Genbank using Batch Entrez.

To select the long-B systems that can be characterized using *E. coli* as a proper host, the protein sequences of og_15, og_44 and og_100 members from the previous study[7] were used as queries to search their homologs in the non-redundant protein sequences database with the NCBI blastp suite. Then, we selected the BPAN system and BPAS system from Gammaproteobacteria, and the BPAM system from *Elizabethkingia anophelis*, a mesophilic pathogen (Fig. 1e and Supplementary Data 1). The gene clusters and neighborhoods of the systems were confirmed manually.

### Phylogeny construction
Homologous sequences were aligned with MAFFT using the automated strategy (v7.490)[57]. Phylogenetic trees were constructed using maximum-likelihood method by FastTree. The results were saved as

newick files and imported into iTOL to plot unrooted tree (v6.5.8; itol.embl.de)[58].

## Tanglegram
The phylogenetic trees of long-B pAgos and og_15, og_44 or og_100 proteins were imported into R environment (v4.2.1). The tanglegrams were visualized using the cophyloplot function of 'ape' R package (v5.6-2)[59] with the association information between long-B pAgos and their respective associated proteins.

## Bioinformatics analysis
The structures of long-B pAgo-associated proteins were predicted by AlpfaFold2[60] using the COSMIC[2] platform (https://cosmic2.sdsc.edu/). Protein sequence alignment of EcbAgaN homologs and GbbAgaS homologs was performed using Clustal W, and the results were visualized with ESPript (https://espript.ibcp.fr/ESPript/ESPript/). Trans-membrane region was predicted with DeepTMHMM (https://dtu.biolib.com/DeepTMHMM).

## Statistics and reproducibility
The cell viability assay, NAD$^+$ level assay and growth curve assay were performed in 3–5 biological replicates as indicated in the figure legends, of which the average values are shown in the graphs. The statistical analyses were performed with Excel. The one-sided Student's $t$-test was used to calculate the $p$-value: $<0.05 = *$; $<0.01 = **$, $<0.001 = ***$.

The correlation of the assigned small RNA sequences and transcriptome RNA sequences was analyzed using Origin 2018, with the Pearson correlation coefficients and corresponding $p$-values calculated.

The experiments related to Figs. 2, 3b, c, 4a, b, 5, Supplementary Figs. 4, 6, 7, 8a, 9a, b, 12, 13, were repeated at least three times independently with similar results.

## Reporting summary
Further information on research design is available in the Nature Portfolio Reporting Summary linked to this article.

## Data availability
All biological materials will be shared by the authors upon request. In addition, small RNA and transcriptome sequencing data are available on the NCBI Sequence Read Archive under BioProject ID PRJNA1003211 (https://www.ncbi.nlm.nih.gov/bioproject/PRJNA1003211). Source data are provided with this paper.

## Code availability
The code used to prepare the figures has been uploaded to https://github.com/yanqiuLiu0908/BPAN-system-analysis.git.[61]

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

## Acknowledgements

The research was supported by the National Key Research and Development Program of China (2022YFA0912200, 2019YFA0906400), National Natural Science Foundation of China (Grant No. 31970545, 32270099, and 31970544), Fundamental Research Funds for Central Universities (Grant No. 2662020SKPY001), the Foundation of Hubei Hongshan Laboratory (No. 2021hszd022) and Huazhong Agricultural University Scientific & Technological Self-innovation Foundation. We thank the core facilities of the Center for Protein Research (CPR) and Experimental Teaching Center of Bioengineering at Huazhong Agricultural University for technical support.

## Author contributions

X.S., S.Lei, S.Liu and Y.L. conducted the experiments with the assistance of P.F., Z.Z., K.Y. and Y.C. Y.L. performed phylogenetic analysis and RNA sequencing analysis. M.L. and Q.S. gave important advice and critically commented on the draft. W.H. acquired the funding, supervised the work and wrote the original draft. All authors contributed to the review and editing.

## Competing interests

The authors declare no competing interests.
