## [Peer Review File · Nature Communications]

Catalytically inactive long prokaryotic Argonaute systems employ distinct effectors to confer immunity via abortive infectionREVIEWER COMMENTS

Reviewer #1 (Remarks to the Author):

This article (Han et al.) is timely and reveals new and relevant information about the diversity of catalytically inactive long-B pAgos, their association with effectors with diverse catalytic activity. The structure of this article (starting with a bioinformatic analysis and then characterizing representatives of the major groups of long-B pAgos) is logical and clear. The results obtained by the authors clearly demonstrate that catalytically inactive long-B pAgos act as modules of foreign genetic information (e.g. plasmids and possibly bacteriophages) that can activate various (nuclease, NADase or transmembrane) effectors, which later using abortive infection kills the infected cell thus preventing the further spread of the infection. This article focuses on long-B pAgo, which is associated with the PD-EXK nuclease domain-containing effector. The results obtained are really convincing. Interestingly, this long-B pAgo is activated specifically by binding transcripts from the pCDF plasmid containing the CloDF13 ori, perhaps due to their abundance. Once the pCDF plasmid is in the cell, EcAgo activates the EcbAgaN effector, which degrades both genomic and plasmid DNA, thereby killing the cell. Other types of long-B pAgos are less well characterized, but they exploit the catalytic (eg NADase) or transmembrane activities of associated effectors to kill cells.

In my opinion, this article can be published in the journal Nature Communication, if the comments below are taken into account.

Major comments:

- 1) In the case of effector protein mutants, appropriate controls must be provided to ensure that the mutations in their active site or deletion of a functional domain have resulted their loss of activity. In the case of in vivo studies, Western blot analysis must be provided showing that the cellular concentrations of both wt and mutant proteins are comparable. In the case of in vitro studies, where preparations of purified proteins are used, CD spectra of both wt and mutants should be recorded to ensure that the mutations have not disrupted the spatial structure of the proteins.
- 2) I suggest using "catalytically inactive pAgos" instead of "inactive pAgos" in the article, because the mentioned pAgos indeed have activity - they recognize the target with the help of a guide and activate effector proteins or domains. I also suggest changing the title of the article to "Catalytically inactive long prokaryotic Argonaute systems employ various effectors to confer immunity via abortive infection".
- 3) EMSA experiments. First of all, if the binding buffer contains Mg²⁺ and/or Mn²⁺, which are important for the protein, TB buffer (Tris-borate) cannot be used as the loading buffer, because both Mg-borate and Mn-borate are insoluble, so during gel loading pAgo will not be able to use these ions for the binding of the 5'-phosphate of the guide, which may distort the experimental results. In this case, Tris-acetate loading buffer must be used, since both Mg-acetate and Mn-acetate are soluble. Second, the experiments shown in Figure 5B use a 5x molar excess of EcAgo over the 5P-RNA guide concentration, i.e. during the binding reaction, 4/5 of the protein remains unbound and could itself bind the labeled target. In order to prove that this scenario does not occur, a control experiment should be performed in which only the protein (without the guide) at the maximum concentration is mixed with the target. Third, since the authors state in Figure 5B that a free guide-target duplex is formed, a control for such a pre-annealed duplex should be provided alongside. Therefore, I suggest conducting EMSA experiments using appropriate experimental conditions with appropriate controls.

Open questions that could be further discussed:

- 1) Line 272: "This reveals that the CloDF13 origin activates the GbBPAS system as observed for the EcBPAN system." Is there any rational explanation why only CloDF13 ori (pCDF plasmid) and not ColE1 ori (pBAD plasmid) activates these systems? Both plasmids are multicopy, and the DNA sequences of their ori regions are very similar, their amplifications mechanisms are also very similar. Does activation depend only on the abundance of plasmid transcripts (seems in the case of the pCDF plasmid there are more of them than in the case of pBAD)?
- 2) Line 278: "The results show that GbAgaS individually and the GbBPAS system together with pCDF-EGFP induce significant culture growth retardation and NAD⁺ level reduction (Figure S7A and B, Figure 6C and D), in line with the cell viability results." The observed decrease in NAD⁺

concentration in the cell is only 2-3 folds, which maybe could hard to explain the approximately 100-fold decrease in cell viability.

3) Line 295: "However, it does not induce significant membrane depolarization or membrane destruction (Figure S7D and E) as other defense systems carrying a TM effector usually do 21,26,35. ". Maybe there are rational explanations for how cells could be killed?

Grammar and others:

- 1) line 42: "all the four" -> "all four";
- 2) line 45: "extrachomosomal" -> "extrachromosomal";
- 3) line 52: "(preduso)" -> "(pseudo)";
- 4) line 94 and elsewhere: "argonaute" -> "Argonaute";
- 5) "in vivo" and "in vitro" should be italic;
- 6) line 282: "Together, The data" -> "Together, the data";
- 7) line 293: "Again, the EaBPAM" -> "Again, the EaBPAM";
- 8) line 299: "three different long-B pAgo systems" -> "three different types of long-B pAgo systems";
- 9) line 449 and elsewhere: when specifying the pH of the buffer, the temperature at which the pH was determined must be specified, as the pH of buffers can be strongly temperature dependent;
- 10) line 463 and elsewhere: "oligos" -> "oligonucleotides";
- 11) line 471 and elsewhere: "laser-scan" -> "laser-scanner";
- 12) line 506: "20 μ M Tris-HCl" -> "20 mM Tris-HCl";
- 13) line 520: "protein expression" -> "gene expression";
- 14) line 617: "Protease K" -> "Proteinase K";
- 15) line 932: "cytomerty" -> "cytometry";
- 16) line 1055 and elsewhere: "pBad24-EcAgo743 or pBad24-EcAgo743-EcbAgaN" -> "pBAD24-EcAgo743 or pBAD24-EcAgo743-EcbAgaN";
- 17) line 1079: "E. DiBAC4" -> "D. DiBAC4".

Good luck,
Mindaugas Zaremba

Reviewer #2 (Remarks to the Author):

Song, Lei, Liu & Liu et al. explore a new bacterial defense mechanism mediated by a group of long inactive prokaryotic Argonaute proteins (long-B pAgos) in conjunction with corresponding effector proteins. In his pioneered work in 2013, Olovnikov et al. demonstrated that a representative long-B pAgo from *Rhodobacter sphaeroides* (RsAgo) uses small 5'-phosphorylated RNA guides to bind complementary DNA targets in its native host and decreases the amount of plasmid DNA and expression of plasmid-encoded genes in a heterologous system. Unfortunately, the underlying molecular mechanism of this phenomenon remained unclear. This manuscript demonstrates that many long-B pAgos act together with associated effector proteins to trigger cell death via various abortive infection (Abi) mechanisms. These results are sound and of great importance to the fast-growing body of scientists studying bacterial immune systems and related areas.

The manuscript is divided into three unequal parts:

1. A short in silico analysis based on the existent data unravels the tight functional connection between long-B pAgos and effector proteins from four orthogroups.
2. The main part where the authors perform detailed biochemical and functional characterization of one such system, BPAN (long-B prokaryotic Argonaute nuclease) system from *Escherichia coli*.
3. Finally, the authors show that members of the other two systems, BPAS (long-B prokaryotic Argonaute Sir2) from *Gammaproteobacteria* bacterium and BPAM (long-B prokaryotic Argonaute trans-membrane) from *Elizabethkingia anopheles*, mediate Abi through NAD⁺ depletion and a yet unknown mechanism involving transmembrane protein effector.

Although the methodological approach chosen by the authors is appropriate and well-established, there are some major concerns in this regard that might affect the proper interpretation and,

therefore, must be clarified. I believe that the manuscript could be suitable for publication in Nature Communications given that the major points listed below are addressed.

Major points:

- The main conclusion of the manuscript, as highlighted in the title, is that long-B pAgos employs various effectors to provide immunity against genetic invaders via abortive infection. Yet, the precise role and the underlying molecular mechanism of pAgos contribution are not clear. For instance, how does EcAgo activate the corresponding nuclease? Do pAgos form complexes with their effectors? In the case of EcAgo the latter question can be addressed by performing *in vivo* and/or *in vitro* pull-down experiments.
- EcbAgaN cleaves both dsDNA and ssDNA targets nonspecifically at 37 C as shown in Figure 2B. What is the size of the short nucleotide products generated by EcbAgaN *in vitro*? Is EcbAgaN an endo- or exonuclease? If it exhibits exonuclease activity, does it have a preference towards 5'- or 3'-end?
- Related to the above question: many non-specific nucleases are very challenging proteins for heterologous expression in *E. coli*. How do authors explain the absence of any apparent negative effect of EcbAgaN expression on cell growth in experiments shown in Figures 2A and 2D?
- Lane 120-122: the authors claim that EcbAgaN efficiently cleaves plasmid and genomic DNA *in vitro* based on the results shown in Figure 2C. What are the molar ratios of plasmid DNA substrates to the enzyme here? It is hard to interpret these data given the low resolution of the image and the absence of the linearized and nicked plasmid controls. While the band distribution pattern changes in the case of a small pUC19 plasmid, it is not obvious for other substrates. One cannot rule out the possibility that EcbAgaN contains a small fraction of contaminating cellular nucleases. Thus, the experiment should be repeated with catalytically inactive single and double mutants of EcbAgaN (similar to Figure 2D in the case of synthetic oligonucleotide substrates).
- Most of the manuscript is dedicated to the study of BPAN system from *E. coli* strain CAP29. The authors identified two potential ORFs for EcAgo encoding 743aa and 731aa long versions respectively. Results of the key *in vivo* experiments shown in Fig.3 were obtained using strains expressing EcAgo743 while most *in vitro* studies were done using EcAgo731, which is confusing and makes the results difficult to explain. Unfortunately, the size exclusion chromatography profile of EcAgo743 shown in Figure S2C displays a sign of massive protein aggregation: a major peak appears right after the column void volume followed by a tail that reaches the peak corresponding to the monomeric version of the protein. Moreover, on lanes 231 – 233 the authors state: "binding by EcAgo743 resulted in that the nucleic acid substrates were stuck in the wells of the gel (data not shown)." These observations clearly indicate that the longer version of the protein is unstable and prone to aggregation which questions the validity of data obtained with EcAgo743. Given the fact that this system comes from a different strain of the most studied model organism – *E. coli*, have authors considered experiments that might shed light on which version of the protein is expressed *in vivo*?
- *In vitro* biochemical characterization of both EcbAgaN and EcAgo lacks rigour. Specifically, optimal pH, ionic strength, temperature, etc. should be determined for any newly purified protein. Of notable concern, the nuclease activity of both EcbAgaN and EcAgo was tested in 20mM Tris-HCl buffer pH 7.5 supplemented with 5mM of each Mg²⁺ and Mn²⁺. Such low ionic strength conditions can promote non-specific binding of DNA/RNA-interacting proteins to nucleic acids and should be generally avoided.
- EcAgo and EcbAgaN are natively organized into a single operon with overlapping ORFs. The authors have chosen to express both components of BPAN from two separate promoters for *in vivo* experiments. In this implementation, it is virtually impossible to control the stoichiometry of two proteins which can lead to unpredictable behaviour.
- The results presented in Figure 3B are hard to interpret due to the apparent unequal loading. The authors should consider some way of normalization: either an equal amount of gDNA per well or

an equal amount of the cells taken for gDNA isolation.

- Lanes 200-201: the authors make a statement about the absolute amount of small RNA guides associated with EcAgo743 in the presence and absence of pCDF-EGFP based on the difference in the signal intensity of the radiolabeled nucleic acids shown in Figure 4A. This is confusing given the fact that the total protein amount taken for nucleic acids extraction was not normalized (based on the information provided in the methods section). Authors should consider performing Western blot analysis to verify equal EcAgo amount between the samples. Additionally, the control sample of radiolabeled total nucleic acids isolated from the IMAC eluate of the lysate of E. coli strain carrying an empty vector will help to verify that the signal indeed comes from the small RNAs associated with EcAgo.

- The authors have chosen not to use gel-purified small RNAs as the starting material to prepare small RNA libraries for high-throughput sequencing. Although this is a valid approach, a control library constructed from nucleic acids isolated from the lysate of E. coli strain carrying an empty vector should be run in parallel for background correction. In addition, how was the size selection step performed? How many cycles of enrichment PCR were run? Did the final library show any sign of overamplification? The final QC report would be helpful here. The same questions apply to the total RNA-seq. Additionally, how the authors performed rRNA depletion?

- Assigned read counts presented in Figure 4C should be at the very least normalized to the library depth for correct comparison. Have the authors considered only single mapped reads for any subsequent analysis shown here? A second biological replicate of small RNA libraries is crucial to confirm that the observed small RNA read length distribution is not a cause of technical variation.

- To increase reproducibility and transparency, all bioinformatics analysis must be well documented, and the code made accessible through one of the many available repositories. Given the information provided in the current manuscript, it is virtually impossible to reproduce the results shown in Figures 4C, 4D, 4E, 4F, S4, and S5.

- RNA and DNA oligonucleotides used as guides to study binding properties of EcAgo (OSX9-12) contain 3'-FAM modification that can negatively affect the binding of the guide 3'-end in the PAZ domain. The authors should consider performing quantitative analysis to calculate the dissociation constant for binary complex formation and compare it to the other known RNA-guided pAgos (RsAgo, MpAgo, etc.). Moreover, these oligonucleotides have 5'-U while HTS data suggest that EcAgo has a bias towards small RNAs with adenine in the 1st position.

Minor points:

- An appropriate measure of confidence, such as pLDDT, must be provided for all Alfold2 predictions.
- All graphs involving flow cytometry data lack the indication of scale on both axes.
- Protein IDs are missing in Figures S1C and S1F.

Reviewer #3 (Remarks to the Author):

Dear editor, dear authors,

In their manuscript, Song et al. have addressed a long-standing question in the fields of Argonaute biology and prokaryotic immunity: What are the function and mechanisms of long-B pAgos.

They combine in vivo and in vitro experiments to reveal the functional and mechanisms of BPAN in detail, while the BPAS studies are mostly limited to convincing in vivo studies, and BPAM studies are very limited and raise more questions than they answer.

This study provides an important advance in the field and most conclusions are nicely supported by the experiments. However, I raise several points that need to be addressed before the manuscript should be considered for publication.

Figure 1: It is unclear to me if this tree contains only (or mostly) those of og_15, 100, 54, 44, or if all long-B pAgos as identified by Ryazansky et al. have been included. It would help if the authors indicate the number of long-B pAgos used, and identified for each subclade. I noticed later that in the discussion it is stated these these systems are representative of 60% of all long-B systems - this is currently not clear from this figure. Furthermore, it would be informative if the authors could explain what other long-B systems exist (i.e. with what other putative domains do they associate?).

Figure 3: Do I understand correctly that the system used by the authors relies on medium-copy number plasmids from which the BPAN system is expressed using T7 RNA polymerase (i.e. heavy overexpression)? This will result in the system being present in extraordinary (unnatural) high quantities. As such, it cannot directly be concluded that the system induces abortive infection, as this might be the consequence of unnaturally high levels of this (activated) system. In fact, the following results showing that ecBPAN can deplete pCDF plasmids could suggest that indeed a more-targeted DNA degradation (also) occurs, and that in fact BPAN interferes with plasmid DNA directly. This does not rule out it can also trigger Abi, but it is not necessarily its only mode of action.

I find the inclusion of both ecAgo731 and EcAgo743 confusing and would suggest to remove one to improve readability.

Figure 4: Although not unlikely, I am not convinced the conclusion that nucleic acids are less abundant in the sample lacking pCDF-EGFP based on a single nucleic acid prep - there are many factors that can affect successful guide purification and/or labelling (line 200). Replicates should be used to verify this and quantification can be done by using ³²P labelling as well as other methods. The same is true for all sequencing experiments, only relative (not absolute) comparisons can and should be made from such sequencing experiments.

Figure 4D: Why are there differences in the observed Pearson correlation coefficients for the different samples? E.g. +bAgaN sample (0.034) shows no correlation.

Line 208: pCDF-EGFP does not generate guides. If so, the authors need to show this. I think they probably mean an unidentified RNase generates guides from the pCDF-EGFP transcripts?

The authors nicely show pCDF triggers activation of the BPAN system. Based on the fact that Agos are activated on a sequence-specific manner, the authors should expand the sequence data analysis to reveal what is the specific pCDF trigger for activation. They do this in the discussion, but this discussion can be further supported by further sequence analysis (and for example showing % of reads mapping to CloDF13 and ColE1 oris).

Line 243: This suggests that the mutation is a recent event which is probably not the case. In addition, the inactivation could rely on other things beyond the mutation. I think the only conclusion that can be drawn here is that the Ago does not show guide-mediated target cleavage under the tested conditions.

Line 250: There is clear background activity of the EcAgo-AcbAgaN complex (i.e. in absence of guides/targets), especially on plasmid DNA. The authors should at least discuss this, and do they perhaps have an explanation for this? Is it possible guides/target copurified? Is it possible the conditions are not suitable for proper control of the system? Is the nuclease present in excess? Does the complex not properly form?

Figure 7: The results for the EaBPAM system are unclear and not fully explored. At this point the EaBPAM system is just toxic when expressed, and why it is toxic is unclear. Therefore, these results currently raise more questions than they answer. I suggest to completely remove this section, unless the authors can expand the characterization by showing what triggers the system

and what causes the toxicity.

Discussion:

-The authors have barely characterized the eaBPAM system, and should rephrase their first part of the discussion.

-I disagree with the conclusions of the EcBPAN system. It might very well be that under natural conditions (i.e. probably lower expression of EcBPAN) it can occasionally result in cleavage of the invading plasmid only. The fact that another plasmid does not trigger the system does not rule that out. Even the current results imply this is not an abortive infection system per-se, but could in some cases clear the invader DNA before triggering abi.

-Line 320: Given the observed background activity of the BPAN system in vitro, the authors should refrain from making any claims on using the system as nucleic acid detection tool (unless they can show it works).

Writing & grammar: While the flow of the manuscript is largely logical, it contains many unclaritys that should be improved to increase accuracy and coherence of the written text. I have given a list of writing errors (non exhaustive) below.

The authors use many distinct terms to indicate the same: Abortive infection, cytotoxicity, cell suicide. This is confusing. The authors would help the readers if give things which are the same, the same name throughout the manuscript.

Introduction

19: catalytically inactive

23: it is unclear here if this system recognizes DNA or RNA?

38: in most PIWI domains (i.e. all short pAgos, all long-B pAgos, various long-A pAgos, and many eAgos) the DEDX tetrad is absent

40: in what sense?

44: ssDNA guides/targets

49: should be emphasized that these preferences are observed in vitro only (i.e. in vivo relevance is unclear)

50: catalytically inactive (i.e. they still bind guides and targets)

51: this suggests the short pAgos also come from Sulfolobus.

56: Is the N-domain fully conserved in long-B pAgos? In certain studies (Ryazanski et al.) it is suggested that it is degenerate. Furthermore, also certain PAZ domains in the long-B clade contain deletions. This should be discussed.

59: in a heterologous host (in which associated gene is absent)

63-64: The types of proteins associated with long-B pagos have been reported in various other studies already (Makarova et al. 2009, Swarts et al., 2014, Ryazanski et al, 2019) and that should be acknowledged accordingly.

Line 220: why is this intriguing? Suggest to refrain from using arbitrary terms.

Line 262: Which asparagine?

Line 269: What plasmid?

I have stopped given textual comments after the introduction, as I do not feel it is my task as a reviewer to enhance writing coherence.

REVIEWER COMMENTS

Reviewer #1 (Remarks to the Author):

This article (Han et al.) is timely and reveals new and relevant information about the diversity of catalytically inactive long-B pAgos, their association with effectors with diverse catalytic activity. The structure of this article (starting with a bioinformatic analysis and then characterizing representatives of the major groups of long-B pAgos) is logical and clear. The results obtained by the authors clearly demonstrate that catalytically inactive long-B pAgos act as modules of foreign genetic information (e.g. plasmids and possibly bacteriophages) that can activate various (nuclease, NADase or transmembrane) effectors, which later using abortive infection kills the infected cell thus preventing the further spread of the infection. This article focuses on long-B pAgo, which is associated with the PD-EXK nuclease domain-containing effector. The results obtained are really convincing. Interestingly, this long-B pAgo is activated specifically by binding transcripts from the pCDF plasmid containing the CloDF13 ori, perhaps due to their abundance. Once the pCDF plasmid is in the cell, EcAgo activates the EcbAgaN effector, which degrades both genomic and plasmid DNA, thereby killing the cell. Other types of long-B pAgos are less well characterized, but they exploit the catalytic (eg NADase) or transmembrane activities of associated effectors to kill cells.

In my opinion, this article can be published in the journal Nature Communication, if the comments below are taken into account.

We thank the reviewer for the positive comments.

Major comments:

1) In the case of effector protein mutants, appropriate controls must be provided to ensure that the mutations in their active site or deletion of a functional domain have resulted their loss of activity. In the case of in vivo studies, Western blot analysis must be provided showing that the cellular concentrations of both wt and mutant proteins are comparable. In the case of in vitro studies, where preparations of purified proteins are used, CD spectra of both wt and mutants should be recorded to ensure that the mutations have not disrupted the spatial structure of the proteins.

Thanks for the suggestions. We have performed western blot analysis of the effectors and their mutants. The results show that the mutations did not affect the stability or the expression level of these proteins (Supplementary Fig. 6).

We have successfully purified the EcbAgaN mutants and revealed that the mutants show the same gel filtration patterns as the wild type protein (Supplementary Fig. 4a), indicating that the mutants also form dimers. We believe that the data could indicate that the mutations have not disrupted the spatial structure of EcbAgaN.

2) I suggest using "catalytically inactive pAgos" instead of "inactive pAgos" in the article, because the mentioned pAgos indeed have activity - they recognize the target with the help of a guide and activate effector proteins or domains. I also suggest changing the title of the article to "Catalytically inactive long prokaryotic Argonaute systems employ various effectors to confer immunity via abortive infection".

Thanks for the suggestion. We have modified the manuscript.

3) EMSA experiments. First of all, if the binding buffer contains Mg²⁺ and/or Mn²⁺, which are important for the protein, TB buffer (Tris-borate) cannot be used as the loading buffer, because both Mg-borate and Mn-borate are insoluble, so during gel loading pAgo will not be able to use these ions for the binding of the 5'-phosphate of the guide, which may distort the experimental results.

In this case, Tris-acetate loading buffer must be used, since both Mg-acetate and Mn-acetate are soluble. Second, the experiments shown in Figure 5B use a 5x molar excess of EcAgo over the 5P-RNA guide concentration, i.e. during the binding reaction, 4/5 of the protein remains unbound and could itself bind the labeled target. In order to prove that this scenario does not occur, a control experiment should be performed in which only the protein (without the guide) at the maximum concentration is mixed with the target. Third, since the authors state in Figure 5B that a free guide-target duplex is formed, a control for such a pre-annealed

duplex should be provided alongside. Therefore, I suggest conducting EMSA experiments under appropriate experimental conditions with appropriate controls.

Thanks for the suggestions. We revised our experimental design in the revised manuscript, including:

(1) We tested different experimental conditions and the chosen buffers are: the EMSA buffer contains 20mM Tris-HCl, pH 7.5, 1mM DTT, 225 mM NaCl and 5mM MgCl₂, while the running buffer is Tris-glycine buffer (25mM Tris, 192mM glycine) with 5mM MgCl₂. The new results are similar to our previous data.

(2) We also analyzed the target-binding of the apo Ago at the maximum concentration. Apo Ago shows a weak affinity to the ssDNA substrates. Supplementation of guide could increase the binding to target ssDNA, but attenuate the binding to non-target ssDNA.

(3) We also included the annealed duplex control in the gels as suggested.

Open questions that could be further discussed:

1) Line 272: "This reveals that the CloDF13 origin activates the GbBPAS system as observed for the EcBPAN system." Is there any rational explanation why only CloDF13 ori (pCDF plasmid) and not ColE1 ori (pBAD plasmid) activates these systems? Both plasmids are multicopy, and the DNA sequences of their ori regions are very similar, their amplification mechanisms are also very similar. Does activation depend only on the abundance of plasmid transcripts (seems in the case of the pCDF plasmid there are more of them than in the case of pBAD)?

Indeed, the amplification mechanisms of the CloDF13 ori and the ColE1 ori are very similar. The only difference between the CloDF13 ori and the ColE1 ori identified in this study is that much more transcripts and small RNAs are generated from the former. Thus, we reason that the abundance of plasmid transcripts determines the ability of CloDF13 to activate GbBPAS and EcBPAN to mediate cell death.

However, our data do not exclude that the ColE1 ori may also activate the EcBPAN system to a lesser extent, which does not induce cell viability reduction in this study.

2) Line 278: "The results show that GbAgaS individually and the GbBPAS system together with pCDF-EGFP induce significant culture growth retardation and NAD⁺ level reduction (Figure S7A and B, Figure 6C and D), in line with the cell viability results." The observed decrease in NAD⁺ concentration in the cell is only 2-3 folds, which maybe could hard to explain the approximately 100-fold decrease in cell viability.

How reduction in NAD⁺ level affects cellular metabolism and cell viability remains unknown. Currently, we cannot explain how the reduction in cell viability is related to the reduction in NAD⁺ level.

We would like to remind that the experimental conditions of the cell viability assay and the NAD⁺ concentration assay are different.

For the cell viability assay, the cells were grown on the plates containing inducers until the survivors are visible for counting (~ 16 h). In comparison, the cells that were subjected to NAD⁺ concentration assay were grown in liquid medium after 2 h' induction of gene expression. We have described the differences in the "**Methods and Materials-cell viability assay**" section.

To avoid any misleading information, we removed "in line with the cell viability results."

3) Line 295: "However, it does not induce significant membrane depolarization or membrane destruction (Figure S7D and E) as other defense systems carrying a TM effector usually do 21,26,35. ". Maybe there are rational explanations for how cells could be killed?

Thanks for the comment. Our current data cannot provide any insights that how BPAM may mediate cell death, except that the TM region of bAgaM is required. We remove these results to avoid raising more questions.

Grammar and others:

- 1) line 42: "all the four" -> "all four";
- 2) line 45: "extrachomosomal" -> "extrachromosomal";
- 3) line 52: "(preduso)" -> "(pseudo)";
- 4) line 94 and elsewhere: "argonaute" -> "Argonaute";
- 5) "in vivo" and "in vitro" should be italic;
- 6) line 282: "Together, The data" -> "Together, the data";
- 7) line 293: "Again, the EaBPAM" -> "Again, the EaBPAM";
- 8) line 299: "three different long-B pAgo systems" -> "three different types of long-B pAgo systems";
- 9) line 449 and elsewhere: when specifying the pH of the buffer, the temperature at which the pH was determined must be specified, as the pH of buffers can be strongly temperature dependent;

Thanks for the remark. We adjusted the pH values of the buffers at 25 °C. We have stated this in the revised manuscript (line 499).

- 10) line 463 and elsewhere: "oligos" -> "oligonucleotides";
- 11) line 471 and elsewhere: "laser-scan" -> "laser-scanner";
- 12) line 506: "20 µM Tris-HCl" -> "20 mM Tris-HCl";
- 13) line 520: "protein expression" -> "gene expression";
- 14) line 617: "Protease K" -> "Proteinase K";
- 15) line 932: "cytomerty" -> "cytometry";
- 16) line 1055 and elsewhere: "pBad24-EcAgo743 or pBad24-EcAgo743-EcbAgaN" -> "pBAD24-EcAgo743 or pBAD24-EcAgo743-EcbAgaN";
- 17) line 1079: "E. DiBAC4" -> "D. DiBAC4".

We thank the reviewer for pointing out the errors. All of them are corrected. We also read through the manuscript carefully and corrected any other errors.

Good luck,
Mindaugas Zaremba

Reviewer #2 (Remarks to the Author):

Song, Lei, Liu & Liu et al. explore a new bacterial defense mechanism mediated by a group of long inactive prokaryotic Argonaute proteins (long-B pAgos) in conjunction with corresponding effector proteins. In his pioneered work in 2013, Olovnikov et al. demonstrated that a representative long-B pAgo from *Rhodobacter sphaeroides* (RsAgo) uses small 5'-phosphorylated RNA guides to bind complementary DNA targets in its native host and decreases the amount of plasmid DNA and expression of plasmid-encoded genes in a heterologous system. Unfortunately, the underlying molecular mechanism of this phenomenon remained unclear. This manuscript demonstrates that many long-B pAgos act together with associated effector proteins to trigger cell death via various abortive infection (Abi) mechanisms. These results are sound and of great importance to the fast-growing body of scientists studying bacterial immune systems and related areas.

The manuscript is divided into three unequal parts:

1. A short in silico analysis based on the existent data unravels the tight functional connection between long-B pAgos and effector proteins from four orthogroups.
2. The main part where the authors perform detailed biochemical and functional characterization of one such system, BPAN (long-B prokaryotic Argonaute nuclease) system from *Escherichia coli*.
3. Finally, the authors show that members of the other two systems, BPAS (long-B prokaryotic Argonaute Sir2) from *Gammaproteobacteria* bacterium and BPAM (long-B prokaryotic Argonaute trans-membrane) from *Elizabethkingia anopheles*, mediate Abi through NAD⁺ depletion and a yet unknown mechanism involving transmembrane protein effector.

Although the methodological approach chosen by the authors is appropriate and well-established, there are some major concerns in this regard that might affect the proper interpretation and, therefore, must be clarified. I believe that the manuscript could be suitable

for publication in Nature Communications given that the major points listed below are addressed.

We thank the reviewer for the comments and important suggestions and questions.

Major points:

- The main conclusion of the manuscript, as highlighted in the title, is that long-B pAgos employs various effectors to provide immunity against genetic invaders via abortive infection. Yet, the precise role and the underlying molecular mechanism of pAgos contribution are not clear. For instance, how does EcAgo activate the corresponding nuclease? Do pAgos form complexes with their effectors? In the case of EcAgo the latter question can be addressed by performing *in vivo* and/or *in vitro* pull-down experiments.

Thanks for the suggestion. We believe that the detailed mechanisms how long-B pAgos activate their effectors should be addressed by extensive structural and biochemical studies, which are beyond the scope of this study.

In the revised manuscript, we analyzed the physical interactions between the pAgos and their effectors via co-expression and pull-down experiments as suggested. The data indicate that only GbAgo forms a stable complex with its effector, while EcAgo or EaAgo does not (Supplementary Fig. 13). The phenomena are discussed in the revised manuscript (line 382-388).

- EcbAgaN cleaves both dsDNA and ssDNA targets nonspecifically at 37 C as shown in Figure 2B. What is the size of the short nucleotide products generated by EcbAgaN *in vitro*? Is EcbAgaN an endo- or exonuclease? If it exhibits exonuclease activity, does it have a preference towards 5'- or 3'-end?

Thanks for the suggestion. The dsDNA and ssDNA degradation assays were performed with size markers in the same gel (Supplementary Fig. 4). This shows that the size of the products is from several nt to ~16 nt.

EcbAgaN efficiently cleaves plasmid DNA, indicating that it is an endonuclease (Fig. 2d). In the revised manuscript, the degradation of both 3'-FAM and 5'-FAM labeled dsDNA and ssDNA was analyzed (Supplementary Fig. 4c). None of the degradation assays generated only 1 nt products, further supporting that EcbAgaN is an endonuclease.

- Related to the above question: many non-specific nucleases are very challenging proteins for heterologous expression in *E. coli*. How do authors explain the absence of any apparent negative effect of EcbAgaN expression on cell growth in experiments shown in Figures 2A and 2D?

In the revised data, we demonstrate that the nuclease activity of EcbAgaN is inhibited by a moderate concentration of NaCl and KCl *in vitro* (Supplementary Fig. 4d), indicating that the EcbAgaN should be suppressed *in vivo*. In addition, we used the Tet promoter to express EcbAgaN in Fig. 3a, which may yield relatively low expression level.

In Fig. 2, we still use the low ionic strength condition to analyze the nuclease activity of EcbAgaN.

- Lane 120-122: the authors claim that EcbAgaN efficiently cleaves plasmid and genomic DNA *in vitro* based on the results shown in Figure 2C. What are the molar ratios of plasmid DNA substrates to the enzyme here? It is hard to interpret these data given the low resolution of the image and the absence of the linearized and nicked plasmid controls. While the band distribution pattern changes in the case of a small pUC19 plasmid, it is not obvious for other substrates. One cannot rule out the possibility that EcbAgaN contains a small fraction of contaminating cellular nucleases. Thus, the experiment should be repeated with catalytically inactive single and double mutants of EcbAgaN (similar to Figure 2D in the case of synthetic oligonucleotide substrates).

Thanks for the remark. We have performed the *in vitro* experiments as suggested, including the linearized and nicked plasmid controls, and the dead EcbAgaN mutant controls. We also calculated the molar concentration of plasmid DNA in the reaction mixture.

In the revised manuscript, we removed the experiments of the degradation of pCDF and pET plasmids, because the yield of the two plasmids was poor.

- Most of the manuscript is dedicated to the study of BPAN system from *E. coli* strain CAP29. The authors identified two potential ORFs for EcAgo encoding 743aa and 731aa long versions respectively. Results of the key *in vivo* experiments shown in Fig.3 were obtained using strains expressing EcAgo743 while most *in vitro* studies were done using EcAgo731, which is confusing and makes the results difficult to explain. Unfortunately, the size exclusion chromatography profile of EcAgo743 shown in Figure S2C displays a sign of massive protein aggregation: a major peak appears right after the column void volume followed by a tail that reaches the peak corresponding to the monomeric version of the protein. Moreover, on lanes 231 – 233 the authors state: “binding by EcAgo743 resulted in that the nucleic acid substrates were stuck in the wells of the gel (data not shown).” These observations clearly indicate that the longer version of the protein is unstable and prone to aggregation which questions the validity of data obtained with EcAgo743. Given the fact that this system comes from a different strain of the most studied model organism – *E. coli*, have authors considered experiments that might shed light on which version of the protein is expressed *in vivo*?

We thank the reviewer for this important suggestion. We predicted the structures of EcAgo743 and EcAgo731, and compared them with the structure of RsAgo. This shows that the N-terminal 12 a.a. of EcAgo743 is disordered and absent from RsAgo, indicating that the EcAgo743 is not the right version.

Thus, we removed the data of EcAgo743 and performed the *in vivo* experiments of the EcAgo731 system (the revised Fig. 3). The results are still consistent with our main conclusion.

- *In vitro* biochemical characterization of both EcbAgaN and EcAgo lacks rigour. Specifically, optimal pH, ionic strength, temperature, etc. should be determined for any newly purified protein. Of notable concern, the nuclease activity of both EcbAgaN and EcAgo was tested in 20mM Tris-HCl buffer pH 7.5 supplemented with 5mM of each Mg²⁺ and Mn²⁺. Such low ionic strength conditions can promote non-specific binding of DNA/RNA-interacting proteins to nucleic acids and should be generally avoided.

We thank the reviewer for this important suggestion. We have screened the optimal *in vitro* experimental conditions with different pH values and NaCl concentrations. The buffer of the EMSA assay was replaced by 20mM Tris-HCl, pH 7.5, 1mM DTT, 225 mM NaCl and 5mM MgCl₂ in the revised manuscript.

In addition, we found that the nuclease activity of EcbAgaN is inhibited by NaCl and KCl *in vitro* (Supplementary Fig. 4d). In the revised manuscript, the low ionic strength condition was still used to analyze the nuclease activity of EcbAgaN (Fig. 2), while a moderate ionic strength (225 mM NaCl) was used in the EcbAgaN activation assay (Fig. 5). We also show that moderate concentrations of NaCl and KCl do not affect the activation of EcbAgaN by the EcAgo-guide-target complex (Supplementary Fig. 12c).

- EcAgo and EcbAgaN are natively organized into a single operon with overlapping ORFs. The authors have chosen to express both components of BPAN from two separate promoters for *in vivo* experiments. In this implementation, it is virtually impossible to control the stoichiometry of two proteins which can lead to unpredictable behaviour.

Thanks for the remark.

Currently most defense systems are characterized in a heterogenous organism. It is always difficult to control the expression of the components to the levels in their native hosts. Even when two components of a defense system are expressed from one promoter, RBS sequence and codon usage bias may also alter their expression levels.

In this study, the *in vivo* characterization of BPAN was performed using different expression strategies, including P_{T7}-Ago and P_{Tet}-bAgaN (Fig. 3a-c), P_{araBAD}-Ago and P_{Tet}-bAgaN (Supplementary 7c) (P: promoter). In the revised manuscript, we also analyzed the phenotypes of the P_{araBAD}-Ago and P_{araBAD}-bAgaN constructions that were constructed with BAC vector (Fig. 3d-e and Supplementary 7a-b), which should yield low and similar expression level of Ago and bAgaN. The results of all the experiments are consistent with our main conclusion.

- The results presented in Figure 3B are hard to interpret due to the apparent unequal loading. The authors should consider some way of normalization: either an equal amount of gDNA per well or an equal amount of the cells taken for gDNA isolation.

Thanks for the suggestion. In the study, the starting OD600 of the cultures of each sample was the same (~0.1) and equal volume (2 mL) of the cultures was used to collect cells and isolate genomic DNA. The same strategy was used in a previous article that characterized another nuclease-based Abi system (David Mayo-Munoz, *Molecular Cell*, 2022).

- Lanes 200-201: the authors make a statement about the absolute amount of small RNA guides associated with EcAgo743 in the presence and absence of pCDF-EGFP based on the difference in the signal intensity of the radiolabeled nucleic acids shown in Figure 4A. This is confusing given the fact that the total protein amount taken for nucleic acids extraction was not normalized (based on the information provided in the methods section). Authors should consider performing Western blot analysis to verify equal EcAgo amount between the samples. Additionally, the control sample of radiolabeled total nucleic acids isolated from the IMAC eluate of the lysate of *E. coli* strain carrying an empty vector will help to verify that the signal indeed comes from the small RNAs associated with EcAgo.

Thanks for the suggestion. We agree that our current data cannot confirm that the amount of small RNA in the absence of pCDF-EGFP is less than that in the presence of pCDF-EGFP. We omit the statement in the revised manuscript. This does not affect the main conclusion of this study.

In addition, we have extracted nucleic acid from the IMAC (immobilized metal affinity chromatography) eluate of the cell lysate of EV, which did not yield any detectable nucleic acid (revised Fig. 4a). The results indicate that the small RNAs extracted from the EcAgo sample are indeed specifically associated with EcAgo.

- The authors have chosen not to use gel-purified small RNAs as the starting material to prepare small RNA libraries for high-throughput sequencing. Although this is a valid approach, a control library constructed from nucleic acids isolated from the lysate of *E. coli* strain carrying an empty vector should be run in parallel for background correction.

As described above, we have extracted nucleic acid from the IMAC eluates of the cell lysate of EV, which did not yield any detectable nucleic acid. Thus, we chose not to sequence the sample, because it has no nucleic acid.

In addition, how was the size selection step performed? How many cycles of enrichment PCR were run? Did the final library show any sign of overamplification? The final QC report would be helpful here. The same questions apply to the total RNA-seq. Additionally, how the authors performed rRNA depletion?

The cycles of the enrichment PCR for small RNA sequencing were 14. The PCR products between 130 and 160 bp were selected for sequencing (corresponding to RNAs between about 10 and 40 nt). Analysis of the PCR products did not detect larger molecular weight products (> 500 bp), indicative of no overamplification.

rRNA was removed by Illumina Ribo-Zero Plus rRNA Depletion Kit (NEB, USA), and we did not remove the rRNA reads when processing the sequencing data.

The above information has been included in the revised manuscript. On the other hand, we think that transcriptome sequencing is a well-established technique, for which it is not necessary to provide a detailed procedure in this manuscript. Other studies did not describe transcriptome sequencing steps in detail either.

- Assigned read counts presented in Figure 4C should be at the very least normalized to the library depth for correct comparison.

Thanks for the remark. In the revised manuscript, we only made relative analysis using the sequencing data as suggested by the third reviewer.

Have the authors considered only single mapped reads for any subsequent analysis shown here?

Only using single mapped reads for the analysis was not adopted by previous studies (Koopal, *Cell*, 2022; Zaremba, *Nature Microbiology*, 2022). We did not use this approach, either.

A second biological replicate of small RNA libraries is crucial to confirm that the observed small RNA read length distribution is not a cause of technical variation.

In the revised manuscript, the small RNA length distributions of three different samples is very similar to each other. Thus, we did not perform a biological replicate to exclude technical variation.

- To increase reproducibility and transparency, all bioinformatics analysis must be well documented, and the code made accessible through one of the many available repositories. Given the information provided in the current manuscript, it is virtually impossible to reproduce the results shown in Figures 4C, 4D, 4E, 4F, S4, and S5.

Thanks for the remark. We have revised the paragraph in the **Methods and Materials** section and uploaded the code used to prepare the figures (<https://github.com/yanqiuLiu0908/BPAN-system-analysis.git>).

- RNA and DNA oligonucleotides used as guides to study binding properties of EcAgo (OSX9-12) contain 3'-FAM modification that can negatively affect the binding of the guide 3'-end in the PAZ domain.

According to the structure of RsAgo, a representative long-B pAgo, the PAZ domain of long-B pAgos is degenerated and does not form a pocket, which is different from typical long-A pAgos. Thus, the 3'-end of the guide is not bound in the PAZ pocket but exposed. Further, 3'-FAM oligonucleotides have been used to analyze the guide affinity of RsAgo (Tomohiro Miyoshi, Nature Communications, 2016). Together, we still used the 3'-FAM guides in the revised manuscript.

The authors should consider performing quantitative analysis to calculate the dissociation constant for binary complex formation and compare it to the other known RNA-guided pAgos (RsAgo, MpAgo, etc.). Moreover, these oligonucleotides have 5'-U while HTS data suggest that EcAgo has a bias towards small RNAs with adenine in the 1st position.

Thanks for the suggestions. In the revised manuscript, the 5'-AU oligos are used as the substrates in the guide-binding EMSA assay and as the guides in the targeting-binding EMSA assay.

The dissociation constant (K_d) values of known pAgos for binding RNA guides are very different, ranging from ~0.05 nM to 50 nM (Zaremba, Nature Microbiology, 2022; Miyoshi, Nature Communications, 2016; Liu, Cell Reports, 2018; Lapinaite, PNAS, 2018). Such differences may partially be resulted from various experimental conditions in the different studies. Thus, we do not think that comparison of the K_d s of different pAgos would provide important insights in this study.

Minor points:

- An appropriate measure of confidence, such as pLDDT, must be provided for all Alfold2 predictions.
- All graphs involving flow cytometry data lack the indication of scale on both axes.
- Protein IDs are missing in Figures S1C and S1F.

We thank the reviewer for pointing out the errors. The essential information, including pLDDT values, scales of flow cytometry data and protein accession numbers, has been provided in the revised manuscript.

Reviewer #3 (Remarks to the Author):

Dear editor, dear authors,

In their manuscript, Song et al. have addressed a long-standing question in the fields of Argonaute biology and prokaryotic immunity: What are the function and mechanisms of long-B pAgos.

They combine in vivo and in vitro experiments to reveal the functional and mechanisms of

BPAN in detail, while the BPAS studies are mostly limited to convincing in vivo studies, and BPAM studies are very limited and raise more questions than they answer.

This study provides an important advance in the field and most conclusions are nicely supported by the experiments. However, I raise several points that need to be addressed before the manuscript should be considered for publication.

We thank the reviewer for the positive comments and important suggestions.

Figure 1: It is unclear to me if this tree contains only (or mostly) those of og_15, 100, 54, 44, or if all long-B pAgos as identified by Ryazansky et al. have been included. It would help if the authors indicate the number of long-B pAgos used, and identified for each subclade. I noticed later that in the discussion it is stated these these systems are representative of 60% of all long-B systems - this is currently not clear from this figure. Furthermore, it would be informative if the authors could explain what other long-B systems exist (i.e. with what other putative domains do they associate?).

Thanks for the suggestions. All long-B pAgos from Ryazansky et al. (199 in total) were included at first; however, only 192 protein sequences were found in NCBI, which were used for subsequent analysis. This has been indicated in the **Method-Phylogenetic analysis** section in the revised manuscript.

The percentage of each subclade has been indicated in Fig .1. Those clustered with og_15, 100, and 44 occupy ~65% of all identified long-B pAgos.

Given the high diversity of long-B pAgo systems, we cannot provide a comprehensive analysis of all of them, which is also beyond the scope of this study. In the revised manuscript, we show representative gene clusters of the long-B pAgos that belong to the og_54, other and none subclades (Supplementary Fig. 1).

Figure 3: Do I understand correctly that the system used by the authors relies on medium-copy number plasmids from which the BPAN systems is expressed using T7 RNA polymerase (i.e. heavy overexpression)? This will result in the system being present in extraordinary (unnatural) high quantities. As such, it cannot directly be concluded that the system induces abortive infection, as this might be the consequence of unnaturally high levels of this (activated) system.

We indeed started with the P_{T7}-expressed EcAgo (Fig. 3a-c), but then analyzed the function of BPAN with lower expression level.

In the revised manuscript, we demonstrate that, when expressed from bacterial artificial chromosome (BAC) plasmid with araBAD promoter, the system can also induce pCDF-dependent genomic DNA degradation and cell death (Fig. 3d, and Supplementary Fig. 7a-b). The data could indicate that the phenotypes are not the consequences of highly-expressed proteins.

In fact, the following results showing that ecBPAN can deplete pCDF plasmids could suggest that indeed a more-targeted DNA degradation (also) occurs, and that in fact BPAN interferes with plasmid DNA directly. This does not rule out it can also trigger Abi, but it is not necessarily its only mode of action.

Thanks for the comment.

Our data reveal that EcBPAN induces a high cell viability reduction on antibiotic-free plates (about 279-fold in Fig. 3a and 46-fold in Fig. 3d), and a low additional cell viability reduction on Str plates (about 13-fold in Fig. 3a and 3-fold in Fig. 3d compared to that on antibiotic-free plates). The results indicate that cell death is the major consequence of the activation of EcBPAN. Further, killing the cells carrying pCDF could also result in depletion of pCDF from the cell population, since the pCDF-free cells have a selective advantage. This is consistent with the immune principle of Abi response. Thus, our data well support that Abi is the main mode of action of EcBPAN.

As the reviewer suggests, we cannot rule out that BPAN cleaves the invader plasmid without degrading genomic DNA in a few cells. This could generate the initial pCDF-free cells. On the other hand, spontaneous plasmid loss may also generate pCDF-free cells. These pCDF-free

cells would finally outnumber the pCDF-carrying cells, due to selective killing of the pCDF-carrying cells. We have revised the Discussion section accordingly (line 331-340).

Please also see our reply to the second question about the Discussion section.

I find the inclusion of both ecAgo731 and EcAgo743 confusing and would suggest to remove one to improve readability.

Thanks for the suggestion. The structural analysis of the two protein versions indicates that EcAgo731 is the right protein. We have performed the detailed in vivo experiments of the EcAgo731 system and removed the results about the EcAgo743 system.

Figure 4: Although not unlikely, I am not convinced the conclusion that nucleic acids are less abundant in the sample lacking pCDF-EGFP based on a single nucleic acid prep - there are many factors that can affect successful guide purification and/or labelling (line 200). Replicates should be used to verify this and quantification can be done by using ³²P labelling as well as other methods. The same is true for all sequencing experiments, only relative (not absolute) comparisons can and should be made from such sequencing experiments.

We agree with the concern of the reviewer and removed the statement that the Ago-associated small RNAs are less abundant in the sample lacking pCDF-EGFP. In addition, we only made relative comparisons using the sequencing data.

Figure 4D: Why are there differences in the observed Pearson correlation coefficients for the different samples? E.g. +bAgaN sample (0.034) shows no correlation.

In the revised manuscript, we analyzed the small RNAs associated with EcAgo731. The abundance of the small RNAs in all samples is well correlated with that of the RNAs from transcriptome.

Line 208: pCDF-EGFP does not generate guides. If so, the authors need to show this. I think they probably mean an unidentified RNase generates guides from the pCDF-EGFP transcripts?

Thanks for pointing out this error. We changed the sentence to "Although the CloDF13 origin is the specific trigger of EcBPAN, only ~1% small RNAs are derived from the CloDF13 origin. This percentage is about 10 times of that of the ColE1 origin (0.11%),"

The authors nicely show pCDF triggers activation of the BPAN system. Based on the fact that Agos are activated on a sequence-specific manner, the authors should expand the sequence data analysis to reveal what is the specific pCDF trigger for activation. They do this in the discussion, but this discussion can be further supported by further sequence analysis (and for example showing % of reads mapping to CloDF13 and ColE1 ori).

Thanks for the remark. We show that the CloDF13 origin of pCDF is the specific trigger with Fig. 3e and Supplementary Fig. 7c. Nevertheless, according to the sequencing data, EcAgo does not selectively acquire guides from CloDF13 ori. In the revised manuscript, we showed the percentages of the small RNAs mapped to different genetic elements (revised Fig. 4 and Supplementary Fig. 8 and 9), which clearly indicate that the small RNAs from the CloDF13 ori are not enriched.

Line 243: This suggests that the mutation is a recent event which is probably not the case. In addition, the inactivation could rely on other things beyond the mutation. I think the only conclusion that can be drawn here is that the Ago does not show guide-mediated target cleavage under the tested conditions.

Thanks for pointing out this error. We changed the sentence to "indicating that EcAgo is not an active DNase."

Line 250: There is clear background activity of the EcAgo-AcbAgaN complex (i.e. in absence of guides/targets), especially on plasmid DNA. The authors should at least discuss this, and do they perhaps have an explanation for this? Is it possible guides/target copurified? Is it possible the conditions are not suitable for proper control of the system? Is the nuclease present in excess? Does the complex not properly form?

In the first version of Fig. 5, EcbAgaN indeed has a strong background activity. In the revised manuscript, we show that the activity is inhibited by moderate ionic strength. In the revised Fig. 5, the moderate ionic strength, which is consistent with the experimental conditions of the EMSA assay, was used. The results do not show any background activity of EcbAgaN or EcAgo-EcbAgaN.

We also demonstrate that in the reaction mixture of the EcbAgaN activation assay, the complex of EcAgo-gRNA-TD indeed formed (Supplementary Fig. 12 a-b).

Figure 7: The results for the EaBPAM system are unclear and not fully explored. At this point the EaBPAM system is just toxic when expressed, and why it is toxic is unclear. Therefore, these results currently raise more questions than they answer. I suggest to completely remove this section, unless the authors can expand the characterization by showing what triggers the system and what causes the toxicity.

Thanks for the remark. We agree that our study on the EaBPAM system is still preliminary. We removed the data that it does not induce membrane depolarization to avoid raising more questions. However, we still decided to keep the results that the system triggers the effector-dependent cell death, because this function is relevant to BPAN and BPAS systems. Further, the data may be helpful for the scientists who are interested in BPAM systems in future.

Discussion:

-The authors have barely characterized the eaBPAM system, and should rephrase their first part of the discussion.

Thanks for the suggestion. We rephrased the sentences to "In this study, we characterized three types of long-B pAgo systems that comprise ~65% of all long-B pAgos (Fig. 1a). The systems are equipped with different associated effectors. Focusing on a long-B pAgo-nuclease (EcBPAN) system, we demonstrate that the system employs the nuclease to mediate genomic DNA degradation and trigger cell death upon recognition of invading nucleic acids, thus defending against the invader via Abi response."

-I disagree with the conclusions of the EcBPAN system. It might very well be that under natural conditions (i.e. probably lower expression of EcBPAN) it can occasionally result in cleavage of the invading plasmid only. The fact that another plasmid does not trigger the system does not rule that out. Even the current results imply this is not an abortive infection system per-se, but could in some cases clear the invader DNA before triggering abi.

Thanks for the comment.

In the revised manuscript, we analyzed the functions of EcBPAN when it is expressed from BAC plasmid using araBCD promoters. The results show that pCDF still activates EcBPAN to mediate extensive genomic DNA degradation and cell death (about 50 folds reduction in cell viability on the antibiotic-free plates), reinforcing that EcBPAN triggers Abi upon activation.

In addition, EcBPAN did not induce depletion of the BAC plasmid after activated (comparison of the cell viability on Chl plates and antibiotic-free plates), indicating that the system does not show biased degradation against extrachromosomal genetic elements.

By comparison, pCDF was selectively depleted from the cell population (about 3 times lower cell viability on Str plates than antibiotic-free plates). The phenomenon could be due to two possible reasons. First, the cells that spontaneously lose the pCDF plasmid can survive and replicate in the presence of EcBPAN, and finally outnumber the cells carrying the pCDF plasmid (Note: the in vivo experiments were performed in the medium without Str). Second, as the reviewer suggested, EcBPAN occasionally cleaves the invading plasmid only, and the resulted pCDF-free cells also have a selective advantage due to the Abi response. Our current data cannot determine which possibility results in the observed pCDF depletion, or both possibilities may simultaneously occur in the cell population. The above discussion has been included in the revised manuscript (line 331-340).

However, we would like to state that the above discussion does not affect our main conclusion that EcBPAN is an Abi system. The conclusion is well supported by the data that pCDF activates EcBPAN to mediate extensive genomic DNA degradation and cell death.

-Line 320: Given the observed background activity of the BPAN system in vitro, the authors should refrain from making any claims on using the system as nucleic acid detection tool (unless they can show it works).

Thanks for the remark. We changed the sentence to "... has potential to be repurposed for sequence-specific applications". Further, we also depleted the background activity with a moderate ionic strength reaction buffer.

Writing & grammar: While the flow of the manuscript is largely logical, it contains many unclaritys that should be improved to increase accuracy and coherence of the written text. I have given a list of writing errors (non exhaustive) below.

The authors use many distinct terms to indicate the same: Abortive infection, cytotoxicity, cell suicide. This is confusing. The authors would help the readers if give things which are the same, the same name throughout the manuscript.

Thanks for the suggestion. We replaced the term cytotoxicity with cell death throughout the manuscript. However, *Abi* refers to a defense strategy via cell death or cell dormancy, the meaning of which is different from cell death or cell suicide.

Introduction

19: catalytically inactive

Corrected as suggested.

23: it is unclear here if this system recognizes DNA or RNA?

Corrected as suggested.

38: in most PIWI domains (i.e. all short pAgos, all long-B pAgos, various long-A pAgos, and many eAgos) the DEDX tetrad is absent

The "PIWI domain" in the context refers to the PIWI domain of eAgos in the context. We added "of eAgos" to avoid any confusion.

40: in what sense?

The bioinformatics studies reveal that eAgos form a single branch on the phylogenetic tree of long pAgos, and that the diversity of pAgos vastly exceeds the diversity of eAgos (Makarova et al. 2009, Swarts et al., 2014, Ryazanski et al, 2018).

44: ssDNA guides/targets

Corrected as suggested.

49: should be emphasized that these preferences are observed in vitro only (i.e. in vivo relevance is unclear)

Modified as suggested.

50: catalytically inactive (i.e. they still bind guides and targets)

Corrected as suggested.

51: this suggests the short pAgos also come from *Sulfolobus*.

We have modified the sentence: "It was reported that short pAgos from many bacteria..."

56: Is the N-domain fully conserved in long-B pAgos? In certain studies (Ryazanski et al.) it is suggested that it is degenerate. Furthermore, also certain PAZ domains in the long-B clade contain deletions. This should be discussed.

According to the structural studies (Tomohiro Miyoshi, 2016; Ryazanski et al, 2018), long-B pAgos have a degenerated PAZ domain, and the conformation of the N-terminal domain is also different from that of typical long-A pAgos. This structural variation may affect the guide and/or target binding behavior of long-B pAgos. We mentioned this in the revised manuscript.

However, this manuscript aims to reveal the immune functions of long-B pAgo systems, instead of the molecular mechanisms how long-B pAgos bind guide and/or target. Thus, we do not discuss the above information in detail.

59: in a heterologous host (in which associated gene is absent)

Modified as suggested.

63-64: The types of proteins associated with long-B pagos have been reported in various other studies already (Makarova et al. 2009, Swarts et al., 2014, Ryazanski et al, 2019) and that should be acknowledged accordingly.

Although Makarova et al. 2009 and Swarts et al., 2014 have revealed the diversity of pAgos and the possible connection between pAgos and their associated proteins, Ryazanski et al, 2018 firstly divided pAgos into three groups (long-A, long-B and short) and comprehensively identified the pAgo-associated proteins. Therefore, we only cited Ryazanski et al, 2018 in this sentence. All the three references have been acknowledged in the previous paragraph.

Line 220: why is this intriguing? Suggest to refrain from using arbitrary terms.

The term has been removed from the revised manuscript.

Line 262: Which asparagine?

The Asparagine (N112) in ThsA has been indicated in the revised manuscript.

Line 269: What plasmid?

The plasmid, pCDF-EGFP, has been indicated in the revised manuscript.

I have stopped given textual comments after the introduction, as I do not feel it is my task as a reviewer to enhance writing coherence.

We thank the reviewer for pointing out the errors. All of them are corrected. We also read through the manuscript carefully and corrected any other errors.

REVIEWER COMMENTS

Reviewer #1 (Remarks to the Author):

The authors performed additional experiments and adequately answered all my questions. The revised manuscript has definitely improved, so I propose to publish it in Nature Communication.

Good luck.
Mindaugas Zaremba

Reviewer #2 (Remarks to the Author):

I thoroughly appreciate the authors' efforts in enhancing the paper's quality. The revised manuscript has undoubtedly shown improvement. The data supporting the mechanism of BPAN-mediated abortive infection via collateral degradation of host genomic DNA are sound. However, I still think that the authors didn't provide enough evidence to claim that long-B pAgos serve as sensors for foreign genetic elements and should adjust the verbatim accordingly. Indeed, HTS analysis of small RNAs associated with EcAgo does not show preferences for CloDF13 origin of replication, and a significant fraction of EcAgo is bound to RNAs derived from normal bacterial transcripts. Moreover, in vitro reconstitution of genomic DNA degradation by EcBPAN clearly demonstrates that this system can be activated by EcAgo loaded with any suitable RNA guide/DNA target pair.

I would also like to ask the authors to clarify the following minor discrepancy that was not fully resolved during the revision process. Specifically, the authors stated that they did not exclude multi-mappers from their bioinformatic analysis because a similar approach was implemented in several other publications. While I value the argument made, I tend to see it differently. I think that the best solution is to choose your own bespoke method(s) that is most appropriate for your experiment and data. Nevertheless, the authors stated that they used FeatureCounts to assign reads to corresponding genetic loci without mentioning any details. If authors used default parameters of FeatureCounts that leads to a contradiction since by default FeatureCounts does not include multimapping reads into downstream analysis.

Finally, I would encourage authors to go through the figures and adjust the font size to enhance readability. One example would be marker labels in Fig.4D.

Reviewer #3 (Remarks to the Author):

Dear editor, dear authors,

The authors have made major improvements in their manuscript and have addressed most of my comments to satisfaction. I have several other comments that can be addressed textually. After addressing these issues, I recommend to accept this manuscript for publication.

I would like to congratulate the authors with their excellent work - it was a pleasure to review this manuscript.

Major issues:

-At this point the authors only show that EeBPAM expression is toxic. (Over)expression of many different proteins is toxic in E. coli. I think it can be concluded that the Ago activates the AgaM protein. However, there is no evidence that it is activated in a guide/target dependent manner, and that it is activated by invading DNA.

As such, it cannot be concluded that it is an abortive infection system. I.e. this system is not characterized, and this conclusion should be removed from the manuscript at all places (abstract, introduction, discussion). Consequentially, the authors cannot state that most long-B Agos act as abortive infection systems (I think it is likely, but there is not enough evidence to support this claim).

If the authors want to keep their data in, they should state something along the lines: as EePAM expression is toxic under all tested conditions, it cannot be concluded whether this system acts as an Abortive infection system. Yet, given its homology to other long-B pAgo systems, and the observation that its expression induces cell death, suggest that also this long-B pAgo might induce Abortive infection akin to BPAN and BPAS systems.

Minor suggestions:

1-I would remove 'various' from the title, or replace it with 'distinct'. Now readers might think a single pAgo employs multiple effectors, for which no evidence is present.

20--The authors do not show that most long-B pAgos act as cell suicide systems. They show it for two systems (see comment above)

67-It would be helpful to indicate that also RsAgo is co-encoded with a nuclease, but that the relevance of that remained unclear.

92-suggests that ... are functionally connected with their respective long-B pAgo (now it suggests that these og are functionally connected)

136/137-I think that this claim should be tuned down (i.e. suggests instead of indicates); in vivo (ionic strength) conditions cannot directly be translated to vitro conditions, as in vivo conidiations might strongly vary locally and might be further influenced by other (bio)molecules. The authors actually show it is inactive a couple of sentences later, so they might instead use this argument at this location to explain why toxicity is observed when only EcbAgaN is expressed.

-270: while this is an important finding, this sentence is not easily comprehensible. Perhaps it can somehow be rephrased?

-There are still several problems with English grammar here and there. Although the manuscript is not incomprehensible, I highly suggest thorough proofreading by the authors, editor, and/or proofreading services.

Reviewer #1 (Remarks to the Author):

The authors performed additional experiments and adequately answered all my questions. The revised manuscript has definitely improved, so I propose to publish it in Nature Communication.

Good luck.

Mindaugas Zaremba

Reviewer #2 (Remarks to the Author):

I thoroughly appreciate the authors' efforts in enhancing the paper's quality. The revised manuscript has undoubtedly shown improvement. The data supporting the mechanism of BPAN-mediated abortive infection via collateral degradation of host genomic DNA are sound. However, I still think that the authors didn't provide enough evidence to claim that long-B pAgos serve as sensors for foreign genetic elements and should adjust the verbatim accordingly. Indeed, HTS analysis of small RNAs associated with EcAgo does not show preferences for CloDF13 origin of replication, and a significant fraction of EcAgo is bound to RNAs derived from normal bacterial transcripts. Moreover, in vitro reconstitution of genomic DNA degradation by EcBPAN clearly demonstrates that this system can be activated by EcAgo loaded with any suitable RNA guide/DNA target pair.

Thanks for the suggestion. We temper the conclusion that long-B pAgos function as sensors in the Abi process (line 22-25, line 331, line 391-392).

I would also like to ask the authors to clarify the following minor discrepancy that was not fully resolved during the revision process. Specifically, the authors stated that they did not exclude multi-mappers from their bioinformatic analysis because a similar approach was implemented in several other publications. While I value the argument made, I tend to see it differently. I think that the best solution is to choose your own bespoke method(s) that is most appropriate for your experiment and data. Nevertheless, the authors stated that they used FeatureCounts to assign reads to corresponding genetic loci without mentioning any details. If authors used default parameters of FeatureCounts that leads to a contradiction since by default FeatureCounts does not include multimapping reads into downstream analysis.

Thanks for the comments. We agree with that our current bioinformatic analysis strategy may result in biased outcomes upon duplicated genes, e.g. *lacI*. The *lacI* gene exists in both the E. coli genome and the pCDF plasmid, and the expression level from the plasmid should be much higher. Thus, the reads from the plasmid *lacI* will be mis-assigned to the genome *lacI*. However, the strategy is well-accepted in this field and the biased results of a few genes do not affect our conclusion. Thus, we still use the bioinformatic analysis strategy in the revised manuscript.

We indeed used the default parameters of featureCounts to assign the reads to genetic loci, which assigns the multimapping reads to each of the multi-copy genes instead of excluding the multimapping reads. The assignment might be random. The

read numbers of each of the multi-copy genes are comparable but not even for unknown reasons.

We have stated the above information in the revised manuscript. (line 757-759)

Finally, I would encourage authors to go through the figures and adjust the font size to enhance readability. One example would be marker labels in Fig.4D.

Thanks for the suggestion. We have enlarged the font size of the small letters in Fig. 4d and other figures.

Reviewer #3 (Remarks to the Author):

Dear editor, dear authors,

The authors have made major improvements in their manuscript and have addressed most of my comments to satisfaction. I have several other comments that can be addressed textually. After addressing these issues, I recommend to accept this manuscript for publication.

I would like to congratulate the authors with their excellent work - it was a pleasure to review this manuscript.

Major issues:

-At this point the authors only show that EeBPAM expression is toxic. (Over)expression of many different proteins is toxic in E. coli. I think it can be concluded that the Ago activates the AgaM protein. However, there is no evidence that it is activated in a guide/target dependent manner, and that it is activated by invading DNA.

As such, it cannot be concluded that it is an abortive infection system. I.e. this system is not characterized, and this conclusion should be removed from the manuscript at all places (abstract, introduction, discussion). Consequentially, the authors cannot state that most long-B Agos act as abortive infection systems (I think it is likely, but there is not enough evidence to support this claim).

If the authors want to keep their data in, they should state something along the lines: as EePAM expression is toxic under all tested conditions, it cannot be concluded whether this system acts as an Abortive infection system. Yet, given its homology to other long-B pAgo systems, and the observation that its expression induces cell death, suggest that also this long-B pAgo might induce Abortive infection akin to BPAN and BPAS systems.

We agree with the comment. We have removed or revised any sentence that states that EaBPAM system is an Abi system. (line 22-27, line 71-75, line 391-394)

Minor suggestions:

1-I would remove 'various' from the title, or replace it with 'distinct'. Now readers might think a single pAgo employs multiple effectors, for which no evidence is present.

The title has been modified as suggested.

20--The authors do not show that most long-B pAgos act as cell suicide systems. They show it for two systems (see comment above)

Thanks for the comment. Please refer to our reply to the major issue.

67-It would be helpful to indicate that also RsAgo is co-encoded with a nuclease, but that the relevance of that remained unclear.

Thanks for the suggestion. RsAgo is associated with a predicted nuclease is mentioned in the revised manuscript. (line 68)

92-suggests that ... are functionally connected with their respective long-B pAgo (now it suggests that these og are functionally connected)

Thanks for the suggestion. We changed the sentence to "... are functionally connected with their respective long-B pAgos." (line 94)

136/137-I think that this claim should be tuned down (i.e. suggests instead of indicates); in vivo (ionic strength) conditions cannot directly be translated to vitro conditions, as in vivo conditions might strongly vary locally and might be further influenced by other (bio)molecules. The authors actually show it is inactive a couple of sentences later, so they might instead use this argument at this location to explain why toxicity is observed when only EcbAgaN is expressed.

Thanks for the suggestion. We moved the argument to the sentences which show that EcbAgaN does not trigger cell death in vivo. (line 151)

-270: while this is an important finding, this sentence is not easily comprehensible. Perhaps it can somehow be rephrased?

Thanks for the suggestion. We have rephrased the sentence: "In addition, the activated DNase activity was not affected by the presence or absence of ~200 mM NaCl or KCl, although the basal activity of EcbAgaN is inhibited by the salts of ~200 mM" (line 271-272).

-There are still several problems with English grammar here and there. Although the manuscript is not incomprehensible, I highly suggest thorough proofreading by the authors, editor, and/or proofreading services.

Thanks for the suggestion. We have proofread the manuscript carefully to avoid any grammar errors.

REVIEWERS' COMMENTS

Reviewer #2 (Remarks to the Author):

The authors have addressed all my concerns and questions. I recommend that the manuscript be accepted for publication in Nature Communications.

Reviewer #2 (Remarks to the Author):

The authors have addressed all my concerns and questions. I recommend that the manuscript be accepted for publication in Nature Communications.

We thank all the comments and suggestions of the reviewers, which have greatly helped us to improve the manuscript.